# Waiting for the better reward: Comparison of delay of gratification in young children across two cultures

**Ning Ding, Anna Frohnwieser, Rachael Miller**©‡*, **Nicola S. Clayton**‡

Department of Psychology, Cambridge University, Cambridge, United Kingdom

‡ These authors contributed equally to this work are joint senior authors.
* rmam3@cam.ac.uk

**Data Availability Statement:** The full data set is available on Figshare: https://figshare.com/s/01356d6162a8b55137c4.

## Abstract

Delay of gratification–a form of self-control–is the ability to forsake immediately available rewards in order to obtain larger-valued outcomes in future, which develops throughout the pre-school years. The majority of previous research in this area has been conducted with Western populations, therefore knowledge of Eastern children's performance is scarcer. Here, utilising on a recently published dataset of British children (n = 61), we further tested delay of gratification in 3 to 5-year-old Chinese children (n = 75) using Bramlett et al.'s (2012) delay choice paradigm. The paradigm was previously used in non-human primates and it featured a mechanized rotating tray that sequentially moves rewards within reach. Additionally, we administered 3 inhibitory control tasks and 1 standardised delay choice task to Chinese pre-schoolers (British children were not tested). We aimed to investigate the influence of culture, reward type and reward visibility on pre-schoolers' ability to delay gratification. We found significant age-related improvements in delay of gratification ability in both countries and children performed better when presented with rewards varying in quality than quantity. Consistent with previous cross-cultural literature, Chinese children showed better overall performance than their British peers when reward visibility was manipulated (though reward visibility itself had no significant effect on performance). There were significant correlations in Chinese children's performance in Bramlett et al.'s (2012) delay choice paradigm and performance in some (though not all tested) inhibitory control tasks. We discuss these results in relation to task demands and the broader social orientation of self-control. We concluded that the intuitive comparative assessment of self-control task taps into children's delay of gratification ability. Our results emphasize the importance of testing for socio-cultural influences on children's cognitive development.

## Introduction

Delay of gratification, a specific form of self-control that involves future-oriented decision-making, is defined as the ability to abstain from taking immediate smaller rewards in order to achieve larger-valued goals in the long-term [1]. From financial decisions to foraging

**Funding:** The study was funded by the European Research Council under the European Union's Seventh Framework Programme (FP7/2007-2013)/ ERC Grant Agreement No. 3399933, awarded to N. S.C (PI), and China Scholarship Council, China, awarded to N.D. The funders had no role in study design, data collection and analysis, decision to publish, or preparation of the manuscript.

**Competing interests:** The authors have declared that no competing interests exist.

behaviours, humans and other animals frequently face inter-temporal choices in which they weigh the costs and benefits associated with immediacy or delayed actions [2, 3]. As a developmental index of self-control [4], young children's propensity to postpone gratifying responses has been associated both with day-to-day functions, such as eating behaviours [5], as well as long-term consequences in academic achievement, social and cognitive competence and well-being [6–9]. With a growing body of animal studies demonstrating its evolutionary origin and significance [10, 11], delay of gratification has become one of the most prolifically researched topics in psychology.

Parallel to the development of many cognitive milestones, children's capacity to delay gratification undergoes dramatic changes in pre-school years and continues to mature into early adolescence [12]. Predominantly, researchers have used two paradigms to measure delayed gratification ability in humans and other animals, specifically delay maintenance and delay choice tasks. In the classic delay maintenance paradigm, the marshmallow test, children were given a single trial to decide whether to have one marshmallow now or wait for more marshmallows later [13–15]. The measure of interest is the length of time lapsed as children need to maintain their action in the face of a tempting treat. On a standardised delay choice paradigm, delay of gratification ability is assessed over multiple trials in which dichotomous choices are made between a smaller immediate reward and a larger delayed reward [16–19].

Researchers have documented pre-schoolers' delayed gratification ability with both primary food rewards, for example, sweets and secondary non-food rewards such as stickers or toys. The typical and consistent findings of delay choice tasks have indicated that 3-year-olds have difficulty in choosing future rewards whereas 4-year-olds and older children demonstrate higher level of success in future-oriented decisions [16–19]. Children in middle childhood continue to show similar age-related performance in delay choice tests. Older children are more likely to delay their gratification, are sensitive to the temporal aspects of different delay options and adopt strategies to decrease their postponed choices with increasing delay interval [20]. With delay maintenance paradigms, researchers have repeatedly found that pre-schoolers' waiting time increases with age. Some previous studies found that a delay above 5 minutes is difficult for 3-year-olds [13, 20], whereas older children can sustain a delay of more than 20 minutes [21, 22]. Recently, emerging evidence has revealed a cohort effect; over the past 50 years, there has been an increase of children's performance in the delay maintenance task, with children born in 2000s waiting on average 2 minutes longer than children in the 1960s [23, 24].

The maturation of delay of gratification ability is a slow process because of its complex cognitive profile. Recent work has identified several potential candidates which scaffold its development in young children. Notably, executive function, a set of higher order cognition, is regularly employed for goal-directed behaviours. Among children, it has been considered as a unitary construct comprising three key components, namely working memory, cognitive flexibility and inhibition [25, 26]. Working memory involves holding and updating information mentally even when it's no longer perceptually present. Cognitive flexibility entails the capacity to switch perspectives and adjust behaviours to changed demands. Inhibition, also referred as inhibitory control, is the ability to control one's attention and behaviour to suppress prepotent impulsive responses and select the more appropriate responses for different circumstances [27]. Although delay of gratification can be considered a form of inhibition, there are fundamental distinctions between self-control and inhibitory control [2]. Specifically, the former requires decisions and actions to sustain a waiting period or to employ greater effort to obtain the delayed but more valuable outcomes, whereas the latter requires suppressing prepotent responses. A battery of tasks, including the Stroop tasks and go/no-go tasks [28, 29], has been developed to measure cognitive and motor inhibition yet not all tasks rely on self-control ability [2].

Theoretically, each executive function component has been hypothesized to contribute differently to children's propensity to resist short-term temptations [30]. The most intuitive link lies between inhibition and delayed gratification, as the former prevents irrelevant thoughts and actions to interfere with future-oriented decisions and to resist the temptation of an immediately available reward [31, 32]. Indeed, better inhibitory control ability has been linked with higher rate of success on delay maintenance paradigms as well as delay choice tasks [33, 34]. Specifically, children who performed better on cognitive inhibition tasks (e.g., Day-Night task) and motor inhibition tasks (e.g., self-ordered pointing test) also showed greater capacity of delayed gratification. Moreover, working memory could facilitate children's ability to postpone gratification by holding the task demand and the goal to obtain a better reward in mind [35, 36]. Neuroscience findings have indicated overlapping prefrontal regions of executive function and inter-temporal decisions [37]. Furthermore, an intervention study, which targeted working memory and inhibitory control, found that through weekly 1.5 hour small group play activities in a school setting for 6–8 weeks, there were significant improvements on children's capacity to delay gratification [38].

Children exhibit varying degrees of delayed gratification ability. These individual differences have been attributed both to their underlying cognitive competency as well as contextual factors, such as reward type and reward visibility. Broadly, researchers have shown that children's performance is influenced by their social perspective; for example, making decisions for another person [19, 39], observing their in-group members choosing to delay [40], perception of the environmental reliability and experimenter trustworthiness [41, 42].

Another line of research has focused on the more fundamental element in delayed gratification scenarios: the reward itself and reward representation. Raclin [43] suggested that reward value varies as a function of quality and quantity, and humans take reward representation into consideration while making inter-temporal choices. Notably, researchers have consistently found that children were more likely to delay when the *quantity* of later options increased [18, 44]. When faced with a significantly larger delayed option, even 2-year-olds could sustain a delay period as long as 16 minutes [45]. Comparably, less is known about the effect of reward *quality* in human developmental literature. In primates, pigs and various avian species, increased delay of gratification performance was found when delayed, higher quality rewards were available in delay maintenance and delay choice task [46–50] as well as in a reverse-reward contingency task [51]. More recently, Miller et al. [52] utilised an intuitive task on delay of gratification, which was originally designed by Bramlett et al. [53] to test self-control in non-human primates, referred to as the "rotating tray" task. Researchers demonstrated that New Caledonian crows and pre-school children showed increased delayed gratification behaviours with qualitatively different rewards than with quantitively different rewards.

More replicated findings come from investigations on reward visibility. Children waited longer for non-visible rewards [13, 54–57]. The strategy of directing attention away from the reward and decreasing its consummatory nature facilitated delayed gratification performance [13, 56, 57]. Such effects also apply to other species as researchers have tested delay of gratification in non-human primates with nonvisible delayed rewards [58, 59]. Specifically, capuchin monkeys exhibited delay of gratification in Bramlett et al.'s [53] rotating tray paradigm even with invisible delayed rewards [59]. Notably, there is a methodological imbalance associated with the findings of contextual factors in the human developmental literature. The majority of studies have adopted the delay maintenance paradigm and the evidence on the role of reward visibility tested with delay choice paradigm is scant [52].

Thus, it would be indecisive and potentially inaccurate to suggest that the contextual factors which influence children's capacity to maintain a delay would work similarly with tasks in which children are required to make dichotomous choices. To date, only two studies have

systematically manipulated reward representation in choice tasks and found mixed findings. First, Garon et al. [44] revealed that children's performance changed as a function of reward quantity but covering the reward had no effect. Second, Addessi et al. [60] found that pre-schoolers displayed greater inclination towards the delayed option with actual food rewards and low-symbolic tokens, but not with more abstract high-symbolic tokens. Given the inconclusive findings, further investigation is required to clarify the role of reward representation on the delay choice task in children.

A noteworthy characteristic of the research on children's delay of gratification is that the data has primarily come from European-American countries, so that our knowledge of its developmental trajectory, correlates and consequences could be culturally skewed and biased [61]. In recent years, there has been increased effort devoted to examining cognitive development outside the Western societies [62]. The development of self-control is acknowledged as a malleable and context-specific process, which is sensitive to social and cultural influences [63]. Each country has its unique set of culturally specific ideologies and social norms; in traditional Chinese Confucian philosophy, self-restraint and inhibition are considered as highly desirable traits and the sign of accomplishment and maturity [64]. Thus, self-control and self-regulation is valued and encouraged more in Chinese society than in Western cultures, which advocate self-expression and assertiveness [65–67]. Consequentially, Chinese parents intentionally adopt specific child-rearing strategies to facilitate self-control behaviours in children and such effort extends to teaching activities and school environment [68–70].

Taking the cultural emphasis and societal effort into consideration, one may expect that Chinese children show advantageous performance on self-control tasks than their European-American peers. Indeed, there are consistent findings suggesting that Chinese children outperform their Western counterparts on a battery of executive function tasks and specifically measures of inhibitory control [71, 72]. Such cultural differences emerge as early as pre-school age and continue into adolescence [73, 74]. However, these studies have predominantly employed cognitive and motor inhibition tasks. There has been little attempt to directly compare children's delay of gratification ability cross-culturally despite its theoretical and developmental significance. To the best of our knowledge, only one study had done so. Lamm and colleagues tested pre-schoolers from a German city and a rural Cameroonian Nso community with a standardised delay choice marshmallow test. The findings indicated that Nso children displayed greater level of delayed gratification ability than their German peers [75]. Socialization goals and parental interactions were correlated with children's delay of gratification performance, a finding consistent with previous literature [76]. Nevertheless, to date there has been no investigation on the Eastern and Western contrast of delayed gratification ability with a delay choice paradigm. Furthermore, the role of reward representation has not been examined in East Asian cultures, posing important questions regarding the universality of contextual factors in children's inter-temporal decisions.

In the current study, we aimed to address these literature gaps by testing Chinese and British pre-schoolers on delay of gratification. Cultural and social environments differ greatly between the United Kingdom and mainland China. Previous researchers have conducted Eastern versus Western comparisons in children's cognitive development with these two populations [73, 77]. In the present study, children's delayed gratification performance was assessed with Bramlett et al.'s [53] delay choice paradigm and we manipulated reward type using rewards differed in quality and quantity. Specifically, the rotating tray task has recently been adopted to test capuchin monkeys [59, 78] as well as comparatively in New Caledonian crows and British children [52].

Furthermore, meta-analysis has revealed a very moderate convergence between the different tests measuring delay of gratification ability [79]. Therefore, in order to compare

performance between the rotating tray delay choice paradigm with several standardised developmental paradigms, we also administered a battery of standardised developmental measures with the same sample of Chinese children (though not British children). Specifically, these were: a standardised delay choice task [19] and three inhibitory control tasks (Day-Night, [80]; Grass-Snow, [81]; Knock-Tap, [82]). Few studies have investigated whether delay of gratification performance is consistent within individuals when tested with different paradigms [52, 78]. Furthermore, to date there has been no investigation dedicated to examining the underlying relationship between the comparative task of delay of gratification [53] and inhibitory control tests among pre-schoolers. In addition to inhibiting the pre-potency of reaching-and-taking responses for the sooner reward, the rotating tray task taps into the ability to make and act upon future-oriented decisions [2, 53]. Therefore, knowledge of how Bramlett et al.'s [53] delay choice paradigm correlate with standardised inhibitory control tasks would shed light on our understanding of the similarities and differences among various measures of self-control.

The overarching aims of our study were twofold: 1) to compare delay of gratification in Chinese and British pre-schoolers, including whether reward representation (Experiment 1) and reward visibility (Experiment 2) influence children's capacity to postpone gratification, and 2) to examine the convergent validity of the rotating tray task by testing whether performance in Bramlett et al.'s [53] paradigm correlates with performance in standardised developmental paradigms in the Chinese sample. Data on the British children came from a recently published study [52] using the same rotating tray task while the Chinese data was collected and utilised only in the current study. Based on existing literature of the Western-Eastern contrast of socialization goals and Chinese children's advantage on various self-control measures, we predicted that Chinese children would outperform their British counterparts in the rotating tray delay choice task. In particular, we hypothesized that Chinese children would make more future-oriented decisions and exhibit greater delayed gratification ability by waiting for the delayed and more preferred rewards. With the rotating tray task, we expected to find universal age-related performance in children that follows a similar developmental trajectory as other delay of gratification paradigms [16–19]. Additionally, Chinese children's performance on Bramlett et al.'s [53] rotating tray task was predicted to correlate with the standardised measures of inhibition and delay choice task. In terms of the contextual factors, we expected to see similar patterns to those in Miller et al. [52], Garon et al. [44] and Purdue et al. [59] where children and non-human primates performed better when the rewards varied in quality than in quantity, and reward visibility had no influence on performance.

## Methods

### Ethics statement

All procedures performed in the present study were in accordance with the ethical standards of and were approved by the University of Cambridge Psychology Research Ethics Committee (PRE. 2018. 080). There was no national regulation applying to foreign researchers and no relevant approval required to conduct data collection in China, however, we did follow the same protocols for the rotating tray task as outlined in our UK ethics approval. Information sheets and consent forms were provided to parents and written parental consent was obtained prior to participation of the children. We also obtained written consent from parents to video record the experimental sessions. For the video in the supporting information, the individual pictured in the video has provided written informed consent (as outlined in PLOS consent form) to publish their image alongside the manuscript.

## Participants

One hundred and thirty-six children aged between three and five-years-old participated in the study. In Britain, 61 children took part in the study: 20 3-year-olds (Mean = 3.65 years, Range = 3.01–3.98 years), 21 4-year-olds (M = 4.68 years, R = 4.05–4.99 years) and 20 5-year-olds (M = 5.34 years, R = 5.05–5.87 years), of which 31 were male and 30 were female. The British participants were recruited and tested at nurseries and schools in Cambridgeshire and Buckinghamshire, which served predominantly white, middle-class backgrounds. In China, we recruited and tested 75 children: 25 3-year-olds (M = 3.65 years, R = 3.02–3.96 years), 25 4-year-olds (M = 4.43 years, R = 4.08–4.95 years), 25 5-year-olds (M = 5.32 years, R = 5.00–5.90 years), of which 40 were male and 35 were female. The Chinese participants were recruited and tested at university affiliated nurseries with Chinese middle-class backgrounds in Kunming, Yunnan Province.

The British data was collected using the rotating tray delay choice paradigm in March-June 2018, with the data set being utilised already in Miller et al. [52] comparative study on delay of gratification in children and New Caledonian crows. The Chinese data (all tasks described in the present study) was collected from November-December 2018 specifically for the present study and has not been used in any other publications to date. Due to the difficulty of tracking previous participants, obtaining consent and the children getting older (which could influence developmental patterns), we were unable to recruit the same group of British children as in Miller et al. [52] again for further testing for the present study. Therefore, we collected and present data in this study on the inhibition control tests and standardised delay choice task from Chinese participants only.

## Apparatus

**Task 1. Rotating tray paradigm (adapted from Bramlett et al. [53] with capuchin monkeys).** In Task 1, the rotating tray paradigm, a 38cm diameter elevated revolving disk was used (Fig 1). The task was adapted from previous non-human primate studies [53, 59, 78]. The disk was mounted on top of a rotation device moving at a speed of 68 seconds per rotation and was operated with a remote control by the experimenter. The apparatus was positioned on a table between the experimenter and the participant, allowing them to sit face-to-face. To prevent children from taking rewards before they were positioned directly in front of them, the revolving disk was contained within a transparent Perspex box (41cm x 34cm x 14cm) with a 29cm x 7cm rectangular opening at the front. We used two small plastic containers to hold rewards (stickers–see details below) and they were positioned at pre-set locations on the disk, referred to as location 1 and location 2. Specifically, the first container at location 1 would

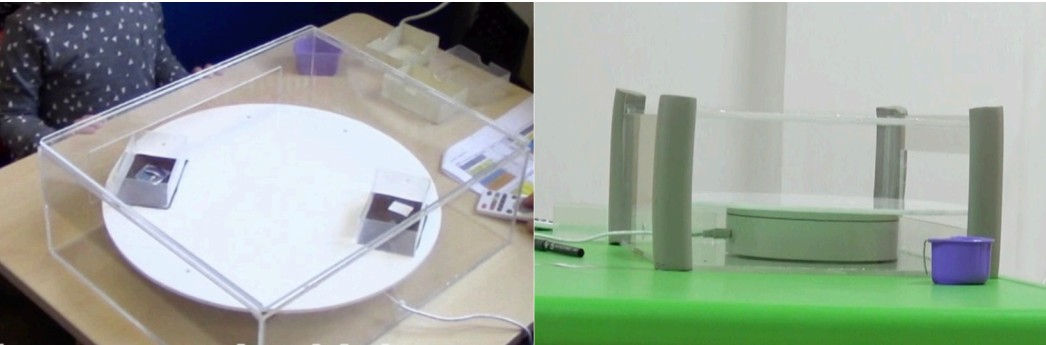

**Fig 1. Rotating tray with two containers.**

come within the participant's reach after 5 seconds, and the second container at location 2 after 15 seconds. These delays were chosen for two reasons; first to be comparable to the previous non-human studies using a similar paradigm [53, 59] and second to be identical to the length of delays adopted in the Miller et al. [52] study with British children.

In the current study, we prepared different types of rewards for the quality and quantity conditions. In the quality condition, the most preferred reward was a large glittery picture sticker (higher quality) and the least preferred reward was a plain sticker of similar size (lower quality). For the quantity condition, we used mini sparkly, picture stickers—the most preferred reward was 4 mini stickers (larger quantity) and the least preferred reward was 1 mini sticker (smaller quantity). Additionally, we also used an "OK" reward which was a yellow smiley sticker during the training in order to maintain participants' motivation during testing as all trials were conducted in one session.

**Preference test.**　To check that the participants were able to accurately select the reward determined to be "most preferred", i.e. higher quality or larger quantity over the reward determined to be "least preferred", i.e. lower quality or smaller quantity, we conducted a preference test between the higher quality sticker and lower quality sticker (quality condition), and for the large and small amount of stickers (quantity condition). We presented both reward options on the table simultaneously and asked participants to pick their favourite option. We ran one trial per condition because our pilot testing revealed that most children showed a clear preference for the most preferred over least preferred options on their first choice. We also wanted to limit the number of stickers being offered in order to maintain motivation in obtaining the rewards across the trials.

**Procedure.**　Participants were tested on a one-to-one basis with a female experimenter in temporary visual isolation from other children in nurseries, preschools and schools (UK) and nurseries (China). In the UK, for some of the younger children, a member of staff was present but did not interact with the participant. To minimise any potential influence of the experimenter and reduce unconscious cueing, we developed a protocol of procedural and specific instructions to guide behaviours during the interactions with participants.

## Experiment 1: The influence of reward type–reward quality vs. quantity

**Training.**　We ran one demonstration trial, where the experimenter started the rotating tray and asked the child to select the container once it arrived in front of them. The experimenter then used two forced-choice training trials i.e. only one transparent container present at either location 1 or 2, with one trial per condition, and no container at the other location, to introduce the rotating apparatus and to ask the participant to select the container with the "OK" reward when it arrived in front of them, within their reach. Participants were then told that there would be 2 containers on the rotating disk, but they could select only 1 container and the disk would stop moving once they made their choice. The purpose of these training trials was to ensure that children were able to pay attention when rewards become accessible and to retrieve the reward in its container from the rotating disk.

**Testing.**　In test trials, the container holding the most preferred reward (higher quality/larger quantity) was placed at location 2 and the least preferred reward (lower quality/smaller quantity) at location 1, and vice versa in control trials. Therefore, in test trials, participants need to wait longer (15 seconds) for the most preferred rewards. The purpose of control trials was to make sure that participants were selecting the most preferred reward on the basis of its location, as opposed to learning to wait for the delayed reward irrespective of the reward type. The rewards were placed inside the transparent containers in sight of participants and they remained visible throughout the trials. We ran separate trials for the quality and quantity

condition. Overall, there were 8 trials: 2 test trials and 2 control trials for each condition and they were administered in a counterbalanced order. Participants were only allowed to make one choice, i.e. to select one container by pointing to it. The experimenter stopped the tray from rotating as soon as a choice was made and children were immediately given their selected reward to keep.

## Experiment 2: The influence of reward visibility

**Training.** The training comprised of 4 forced-choice trials, which was considered to represent a memory test. The purpose of the memory test was to ensure that children were able to remember the location of a hidden reward, so that failure in the testing was unlikely to be attributed to any memory constraints. The experimenter simultaneously picked up the two transparent containers and placed them on the rotating disk. An "OK" reward was placed in one of the containers (location 1 or 2) in sight of participants, with no reward in the second location, and an opaque lid was then placed on both containers. As such, the reward hidden within one of the containers was not visible once baiting had finished and the disk started rotating. In total, there were 4 memory test trials, with 2 trials with the container at location 1 baited with the reward, and 2 trials with the container at location 2 baited. We ran 2 additional trials for children who failed any of the first 4 trials, with one trial per location. All participants passed the memory test; 84% of participants passed in the first 4 trials and 16% of participants scored 3 out 4 in the first 4 trials and passed the subsequent additional trials.

**Testing.** Similar to Experiment 1, we ran test trials with the most preferred reward (higher quality/larger quantity) in location 2 and least preferred reward (lower quality/smaller quantity) in location 1, and vice versa for control trials. The rewards were baited in containers at both locations simultaneously in view of the participants. To investigate the influence of reward visibility, we ran three different test types (Fig 2). In test type 1 (immediate reward visible), the container in location 1 had a transparent lid whereas the container in location 2 had an opaque lid. Therefore, in this case, only the immediately available reward at location 1 was

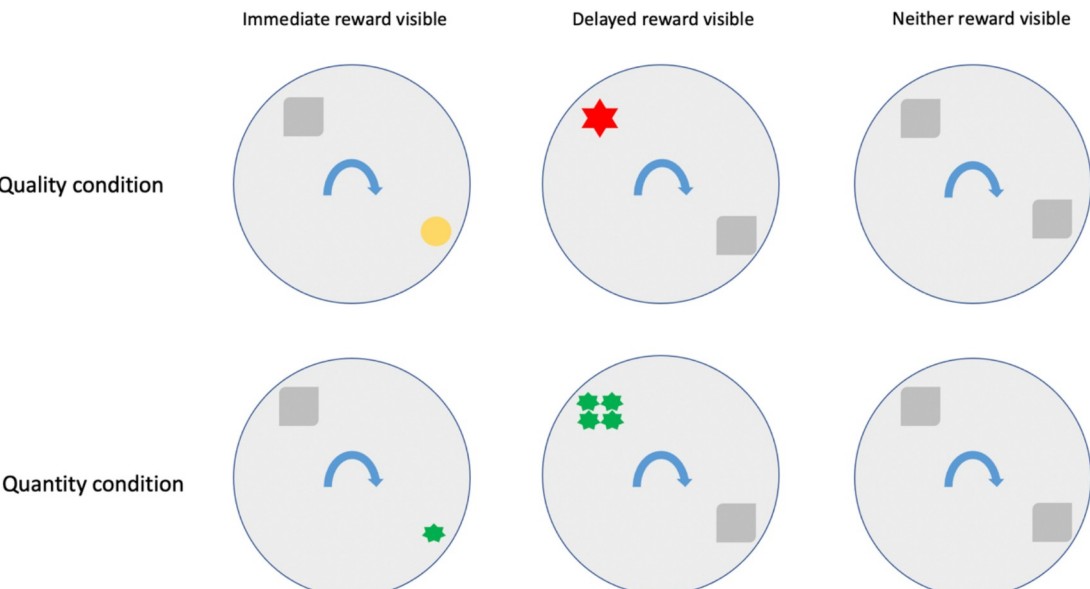

**Fig 2. The quality and quantity condition test trials in Experiment 2.** The least preferred (LP) in location 1 would come within participants' reach after 5 seconds, and location 2 with the most preferred (MP) reward after 15 seconds.

visible once the disk rotation started. In test type 2 (delayed reward visible), only the delayed reward was visible as the container in location 2 had a transparent lid while the container in location 1 had an opaque lid. In test type 3 (neither reward visible), neither rewards were visible once baiting completed as we used opaque lids for both containers.

In Experiment 2, we ran 12 trials with 6 trials for quality condition and 6 trials for quantity condition. Of the 12 trials, we ran 4 trials for each test type and within each test type there were 2 test and 2 control trials. The trials were administered in a counterbalanced order. Children were not allowed to revoke their decision once they selected the container. All participants completed both experiments with a fixed order (Experiment 1 first followed by Experiment 2) within the same session. Overall, the study lasted approximately 20 minutes with a total of 28–30 trials (number of trials dependent on training performance). We randomly assigned participants to 2 subgroups for order of test conditions; half of children received the quantity condition then the quality condition, and the other half received the quality condition first followed by the quantity condition.

## Task 2–5. Standardised developmental paradigms

In addition to Bramlett et al.'s [53] rotating tray paradigm, we administered three standardised inhibition tasks and one delay choice task [19] to the same sample of Chinese pre-schoolers. These tasks were selected on the basis that they were sensitive to detect age-related changes and were widely used in developmental literature [83, 84]. Participants completed these tasks in a fixed order after the rotating tray task and the total length of these tasks was around 8 minutes.

**Day-night [80].**   This was a stroop-like verbal cognitive inhibition task. Children were firstly asked to identify the day (picture of a sun) and night card (picture of a moon). Then they were asked to say "day" for the night card and "night" for the day card as the cards were shown one at a time. There were two practice trials with one trial of each picture card to ensure that children understood the rule. If failed, the experimenter would repeat the instructions and run two more practice trials. 16 cards were presented in a fixed random order and accuracy out of 16 trials was noted.

**Grass-snow [81].**   This was a variant version of the Day-Night task. Children were firstly asked to name the colour of grass and snow. Then the experimenter introduced one green paper and one white paper and asked children to point to the green paper when they heard the word "Snow" and to the white paper when "Grass" was spoken. Children received two practice trials and repeated instructions if necessary. Accuracy out of 16 trials was recorded.

**Knock-tap [82].**   This motor inhibition task measured children's ability to inhibit established motor movement and impulses evoked by visual stimuli. The first part of the task involved children mimicking the experimenter's hand movement. After passing 8 consecutive trials, children were then asked to perform the opposite hand movement from the experimenter. For example, if the experimenter knocks on the table, then the child should tap the table with a flat palm, and vice visa. Accuracy out of 15 trials was recorded.

**Delay choice task [19].**   This was a standardised delay choice paradigm assessing young children's delay of gratification. Children were asked to choose between an immediately available reward of small quantity and a delayed reward of larger quantity. A total of nine trials were administered; created by crossing three types of reward (stickers, animal erasers, cartoon stamps) with three types of choices (1 now vs. 2 later, 1 now vs. 4 later, and 1 now vs. 6 later). Both options were simultaneously and physically presented to children on a table. Children received the immediate reward if they chose it, and the delayed reward was placed in an envelope and remained inaccessible until the end of the study. Children waited approximately 2

minutes and the delay length was the same for all participants. Each trial was solved correctly when the child selected the delayed larger reward. Accuracy out of 9 trials were recorded. This delay choice task was selected because it was a widely used standardised developmental paradigm involving a series of dichotomous choice between immediate and delayed rewards, which was similar to the rotating tray task.

**Task 1: Data analysis for rotating tray paradigm.** We recorded the choice per trial for each child as "correct" or "incorrect", with the correct choice being the reward of higher quality or larger quantity, whether it was immediate (control trial) or delayed (test trial). In the preference trials, 98.6% of British children and 97.3% of Chinese children selected correctly for quality condition and 96.7% of British children and 93.3% of Chinese children selected correctly for quantity condition. If a child failed to select correctly in the initial preference trials, they were administered with one follow-up trial for the specific condition that they failed—the pass rate for the follow-up trial was 100%. Given the methodological and analytical importance of test trials, we present the results of test trials in the manuscript and include the analyses for the test and control trials combined in the supporting information. We live coded as well as video recorded all experimental sessions unless parents requested no recording. Cohen's Kappa was run to test for inter-rater reliability and there was good agreement, κ = .828, p < .001. We aimed to investigate the general developmental trajectory and potential factors affecting children's performance in Bramlett et al.'s [53] rotating tray task across both countries (British, Chinese). With Chinese pre-schoolers, we also aimed to explore whether performance in the rotating tray task correlated with the standardised inhibitory control tests and delay choice task.

Generalized Linear Mixed Models (GLMM: Baayen 2008) in R (version 3.4.3; R Core Team, 2014) were used to assess which factors influenced children's performance in terms of success rate in the rotating tray task. Success rate was the dependent variable in the models and was a binary variable indicating whether the child chose correctly (1) or not (0). In Experiment 1, the random effect included in the models were participant ID and fixed effects included age in years (categorical, ages 3–5 in individual years), country (Britain vs. China), the interaction effect of country and age (Britain vs. China and 3 to 5 years), condition (quality, quantity), order (quality-quantity, quantity-quality) and sex (male, female). For Experiment 2, we included the same fixed effects as Experiment 1 as well as adding the fixed effects of visibility (immediate reward visible, delayed reward visible, neither reward visible).

We used likelihood ratio tests to compare the full models (all predictor variables, random effects and control variables) firstly with a null model, and then with reduced models to test each of the effects of interest [85]. The null models contained the random effects and control variables (i.e. no predictor variables) namely "sex" as we did not predict this variable to significantly effect performance. The reduced models comprised of all effects present in the full model, except the effect of interest. For the GLMMs, we used family = binomial, R package "lme4", "glmer" and "anova" functions [86]. We compared the log likelihood ratio of a) the full with null model, and b) the full with reduced model containing only the main effects to test the effect of the interaction term, and c) the final with reduced models to test each of the effects of interest, using maximum likelihood. The p-values in the models were derived from the likelihood ratio tests. We include a copy of our R script and accompanying data sets for the analysis for Experiment 1 and 2 (https://figshare.com/s/01356d6162a8b55137c4). We ran further analyses for the significant variables identified in the GLMMs where applicable, using Tukey contrasts for pairwise comparisons of age in R, and to compare performance against chance using non-parametric two-tailed statistics (Wilcoxon signed ranks and Mann Whitney U tests in SPSS, version 27).

**Task 2–5: Data analysis for standardised developmental paradigms.** We recorded the total number of correct trials for each standardised inhibition task and the delay choice task.

The scoring methods have been validated in previous research and were consistent with the vast majority of developmental studies [83, 84]. Specifically, for the scope of the present study, the use of sum scores of the standardised delay choice task were appropriate and sufficient in testing age-related performance as well as for correlational analysis. As the data violated the assumption of normality, we used non-parametric Kruskal-Wallis tests to investigate the effect of age on children's performance. Furthermore, Spearman's rank order correlations were conducted to test the relationship among the different tasks. Notably, both the standardised delay choice task and the quantity condition in Experiment 1 of the rotating tray paradigm used quantity rewards that were visible to the participants, therefore we included both the overall performance in the rotating tray task as well as performance in the quantity condition in Experiment 1 for the correlation analysis.

## Results

### Task 1: Rotating tray paradigm–Experiment 1

The full model differed significantly from the null model (Chisq = 39.46, df = 7, $p <$. 001). The full model (with interaction term) was not significantly different to the reduced model (main effects only; Chisq = 4.8, df = 2, $p$ = 0.091). Therefore the interaction term (Age: Country) did not significantly improve the model and the final model reported is the best fit (Table 1). We found a significant main effect of **condition** (quality vs. quantity; Fig 3), **order** (quality-quantity vs. quantity-quality) and **age** (3 to 5 years; Fig 4), with no significant effect of country or sex (Table 1).

Notably, across both countries and all age groups, children's success rate was significantly higher in the quality than in the quantity condition, with above chance level performance within each condition (Wilcoxon signed ranks test: all $p <$ .001). In terms of age, 5-year-olds significantly outperformed 3 and 4-year-olds, with no difference between 4 and 3-year olds (Tukey contrasts: age 5 vs age 4, z = 2.25, p = .024; age 5 vs age 3, z = 3.18, p = .001; age 4 vs age 3, z = 1.04, p = .298), though all three age groups performed above chance (Wilcoxon signed ranks test: Age 3: $p$ = .026; Age 4: = .009; Age 5: $<$ .001). With regard to the main effect of order, children performed better when tested with the order of quality-quantity than quantity-quality with performance significantly above chance in both order groups (Wilcoxon signed ranks test: $p <$ .001). Results using test and control trials combined dataset were presented in the supporting information (S1 Table).

### Task 1: Rotating tray paradigm—Experiment 2

The full model differed significantly from the null model (Chisq = 76.26, df = 9, $p <$. 001). Moreover, the full model (with interaction term) was significantly different to the reduced

**Table 1. Generalized linear mixed models for Experiment 1.**

| Fixed Term | chi-square | df | p-value |
|---|---|---|---|
| Country | 0.06 | 1 | 0.805 |
| Condition | 15.84 | 1 | **<0.001** |
| Age in Years | 9.14 | 2 | **0.01** |
| Order | 8.72 | 1 | **0.003** |
| Sex | 0.27 | 1 | 0.604 |

Generalized linear mixed models (final model) on factors affecting the number of correct *test* trials in children for Experiment 1 with British and Chinese dataset combined. N = Britain 61; China 75. P-values < 0.05 are highlighted in bold. The British data has been published in Miller et al. [52].

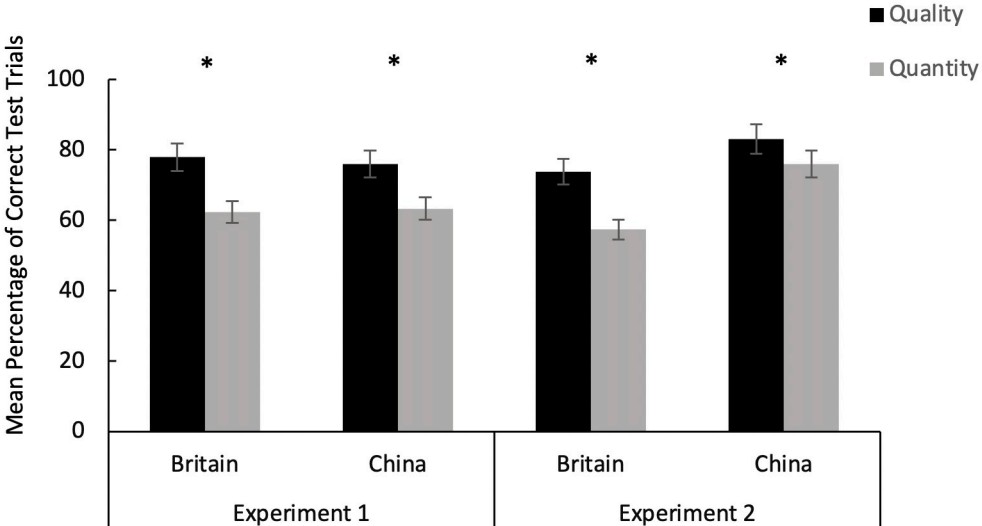

**Fig 3. Mean percentage of correct test trials across age groups by condition (quality and quantity) in Experiment 1 & 2.** * indicates performance above chance level (p < 0.05), error bars indicate standard errors. Data on the British children has been published in Miller et al. [52].

model (main effects only), and is the best fit model (Chisq = 25.303, df = 2, $p < .001$) (AIC equal between models with main effects of age and country included or removed, hence removed for final model). Specifically, there was a significant main effect of **condition** (quality vs. quantity, Fig 3) and an interaction effect of **country** and **age** (Table 2, Fig 5). There was no significant effect of sex, order or visibility (Table 2). Results using test and control trials combined dataset were presented in the supporting information (S2 Table).

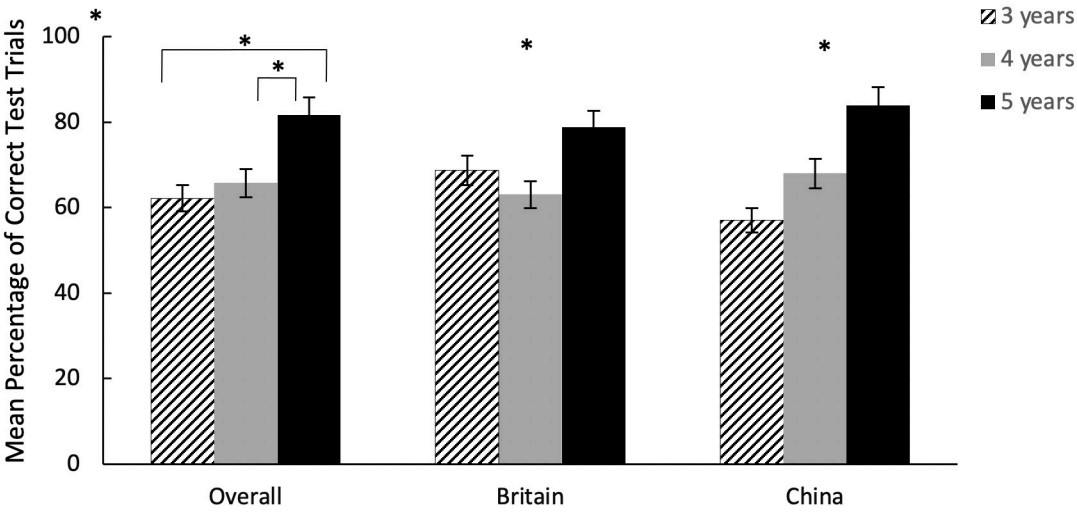

**Fig 4. Mean percentage of correct test trials across conditions by age groups in Experiment 1.** * indicates performance above chance level (p < 0.05), error bars indicate standard errors. Across countries, 5-year-olds outperformed 3- and 4-year-olds respectively, with no difference between 3- and 4- year-olds' performance, indicated by the significance lines. Data on the British children has been published in Miller et al. [52].

**Table 2. Generalized linear mixed models for Experiment 2.**

| Fixed Term | chi-square | df | p-value |
|---|---|---|---|
| Condition | 16.59 | 1 | **<0.001** |
| Country: Age | 57.69 | 5 | **<0.001** |
| Order | 0.146 | 1 | 0.702 |
| Sex | 0.221 | 1 | 0.513 |
| Visibility | 3.12 | 2 | 0.21 |

Generalized linear mixed models (final model) on factors affecting the number of correct *test* trials in children for Experiment 2 British with Chinese dataset combined. N = Britain 61; China 75. P-values <0.05 are highlighted in bold. The British data has been published in Miller et al. [52].

Similar to Experiment 1, in Experiment 2, children's performance was better in the quality than in the quantity conditions with above chance level performance in both conditions (Wilcoxon signed ranks test: all $p < .001$). With the age and country interaction, Chinese 4- and 5-year-old children outperformed their British peers respectively, with no difference in performance between the Chinese and British 3-year-olds. Within country, for Chinese children, 4- and 5-year-olds performed significantly better than the 3-year-olds, with no significant difference between the 4- and 5-year-olds (Tukey contrasts: age 5 vs age 4, z = 0.755, p = .45; age 5 vs age 3, z = 3.59, p < 0.001; age 4 vs age 3, z = 3.002, p = .003). Comparing performance against chance level, only 4 and 5-year-old Chinese children scored significantly above chance with 3-year-olds showing below chance success rate (Wilcoxon signed ranks test: Age 3: $p = .091$; Age 4: $p < .001$; Age 5: $p < .001$). In comparison, British 5-year-olds performed significantly better than 4-year-olds, with no difference between 5 and 3-year olds, or between 4 and 3-year olds (Tukey contrasts: age 5 vs age 4, z = 2.68, p = 0.007; age 5 vs age 3, z = 1.495,

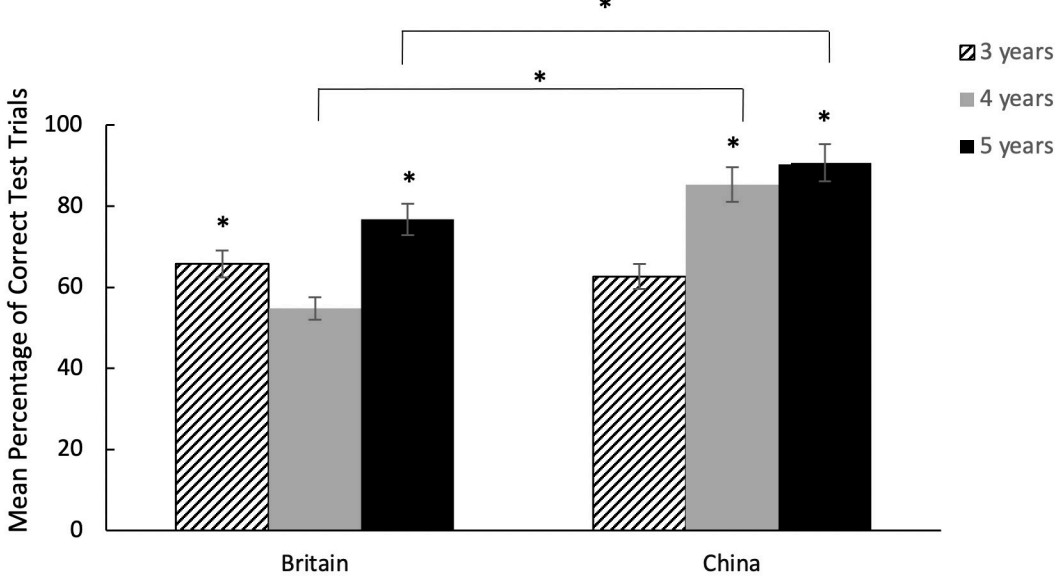

**Fig 5. Mean percentage of correct test trials across condition by age groups in Experiment 2.** * indicates performance above chance level (p< 0.05), error bars indicate standard errors. Chinese 4- and 5-year-olds outperformed British 4- and 5-year-olds respectively, with no differences in performance between the 3-year-olds, indicated by the significance lines. Data on the British children has been published in Miller et al. [52].

**Table 3. Age effect on inhibitory control and delay choice tasks.**

| Task | $X^2$ | Total N | Degree of Freedom | P-value |
|------|-------|---------|-------------------|---------|
| Knock-Tap | 26.269 | 74 | 2 | $< .001$ |
| Day-Night | 9.122 | 73 | 2 | **.010** |
| Grass-Snow | 14.694 | 72 | 2 | **.001** |
| Delay Choice Task | 7.948 | 75 | 2 | **.019** |

Kruskal-Wallis tests on the age effect on inhibition tasks and standardised delay choice task.

P-values <0.05 are highlighted in bold.

p = .135; age 4 vs age 3, z = -1.184, p = .236), with only 3- and 5-year-olds scoring significantly above chance (Wilcoxon signed ranks test: Age 3: $p$ = .011; Age 4: $p$ = .625; Age 5: $p < .001$).

We note that in the 4-year-olds British age group (total $N$ = 21), three children scored zero in the test trials in Experiment 2. In comparison, not a single participant in the 3-year-olds age group failed all test trials. After removing the three participant outliers, additional analysis revealed that British 4-year-olds' success rate was above chance level (Wilcoxon signed ranks test: $p < .001$). Moreover, there was still no difference between British 3 and 4-year olds' performance on the rotating tray task (Mann Whitney U test: $Z^2$ = -1.769, $p$ = .077). We conducted additional analyses using the dataset without the three British outliers. For the GLMMs, there was a significant main effect of condition and an interaction effect of country and age (S3 Table). Further, as with the dataset including outliers, Chinese 4- and 5-year-old children outperformed their British peers, with no difference in performance between the Chinese and British 3-year-olds, when outliers were removed for analysis. Reward visibility alone did not influence children's performance in the test trials on the rotating tray task.

## Task 2–5: Standardized developmental paradigms (Chinese sample only)

Across all standardised developmental tasks (three inhibitory control tasks and one delay choice task), Kruskal-Wallis tests revealed that age had a significant effect on performance (Table 3). Specifically, 5-year-olds outperformed 3-year-olds on all tasks except on the Day-Night task (Mann Whitney U test: all p <. 01). Moreover, 4-year-olds scored significantly higher on the Knock-Tap, Day-Night and Grass-Snow task than children aged 3-years-old (all p <. 02). To investigate the inter-task relationships, we ran Spearman's rank order correlations. We found significant correlations between the inhibition tasks and standardised delay choice task (Table 4).

Notably, a novel finding in our study was that there was a significant correlation between the Knock-Tap task performance and the overall performance in the rotating tray paradigm (Table 4). Additionally, we found that children's performance in the quantity condition in Experiment 1 in Bramlett et al.'s [53] rotating tray task was significantly correlated with performance in the standardised delay choice task—both tasks involving choices relating to reward quantity (Table 4).

## Discussion

In the present study, we investigated Chinese and British pre-schoolers' ability to delay gratification, i.e. inhibit immediate desires for delayed larger-valued goals. The British dataset has been collected and published in Miller et al. [52], whereas the Chinese data were collected for the present study. We used a 'rotating tray' task originally designed to test non-human primates [53], which allowed for systematic manipulations of reward type and visibility, and also

**Table 4. Correlations and descriptive statistics for inhibition control and delay choice tasks.**

|  | 1 | 2 | 3 | 4 | 5 | 6 |
|---|---|---|---|---|---|---|
| 1. Knock-Tap | - | .274* | .378** | .392** | .249* | .113 |
| 2. Day- Night |  | - | .449** | .243* | .038 | .078 |
| 3. Grass-Snow |  |  | - | .336* | .104 | .047 |
| 4. Delay Choice Task |  |  |  | - | .193 | .239* |
| 5. Experiment 1 & 2 Test Total |  |  |  |  | - | .770** |
| 6. Experiment 1 Quantity Test |  |  |  |  |  | - |
| Mean | 12.79 | 11.67 | 12.09 | 6.87 | 7.66 | 1.29 |
| SD | 2.56 | 2.99 | 3.44 | 2.32 | 2.63 | .82 |
| Range | 2–15 | 4–16 | 2–15 | 1–9 | 0–10 | 0–2 |

Correlations and Descriptive Statistics for inhibition control and standardised delay choice tasks. Note.

**p < .01.

*p < .05. Spearman's rank order correlations across age groups are presented.

tested the Chinese children with several standardised developmental inhibitory control and delay choice tasks. Our results add to existing cross-cultural literature on cognitive development, revealing that Chinese children outperformed British peers in the rotating tray delay choice paradigm, though children in both countries performed above chance, when reward visibility was also manipulated. We found age-related performance overall all tasks, with a significant age effect in Experiment 1 and a significant interaction effect of age and country in Experiment 2. Pre-schoolers from both countries exhibited higher delayed gratification performance when rewards differed in quality over quantity, whereas occlusion of rewards had no significant effect. Furthermore, there were significant correlations between performance in the rotating tray task with a standardised motor inhibition task (Knock-Tap) and a delay choice task (in Chinese pre-schoolers only as British children were not tested on the standardised developmental tasks).

With the rotating tray paradigm, the significant age effect in Experiment 1 and interaction between age and country in Experiment 2 was consistent with previous findings employing standardised delay choice and delay maintenance paradigms measuring children's delay of gratification, including the marshmallow test [13–19]. An increased ability in older children to delay gratification suggests that they may be more future-oriented and better at understanding the connection between their present actions and future outcomes [16]. However, even the youngest children tested performed well as reflected in the high rate of above-chance performance in all age groups (Experiment 1 and 2). The short delay (15 seconds) between immediate and delayed rewards may have contributed to this finding by reducing necessary effort to wait. Future studies could increase this delay time to explore the effects on performance across ages. There were two cases where children's performance fell below chance level. First, in Experiment 2 where visual occlusion was manipulated, 3-year-old Chinese children scored below chance level, which may indicate responses based on guesses attributed to task difficulty and cognitive immaturity. Second, in Experiment 2, British 4-year olds' performance was below chance while the 3-year-olds scored above chance level. We detected three outliers in the British 4-year-old group who scored zero in all test trials, in comparison, all 3-year-olds British children scored at least 1 or above. Although removal of these outliers increased British 4-year-olds' performance to above chance level, it did not change the overall findings otherwise (S3 Table).

We found a significant age and country interaction effect in Experiment 2, when the reward visibility was manipulated, though not in Experiment 1, when both rewards were visible.

Specifically, Chinese 4- and 5-year-olds outperformed British 4- and 5-year-olds on the rotating tray task, though we note that children from both countries performed better than chance level. Our findings were in line with previous research on Eastern children's outperformance on self-control [71–74]. It also supports the early prevalence of cross-cultural differences, as socialization and child-rearing strategies towards behavioural and emotional regulation are present for young children in China [87]. At a broader societal level, the cultural differences are likely to reflect consequences of an emphasis on self-control and parental expectations of impulsive control and willpower in China, given the social pressure to compete for higher education resources [88].

The significant effect of condition across age groups and countries (Experiment 1 and 2) indicates that children made more future-oriented decisions when reward differed in quality over quantity. This finding was in line with our predictions and previous related findings with British pre-schoolers [52]. This effect of reward type could be attributed to reward properties and task format. First, it was unlikely that children selected on the basis of which condition was associated with gaining rewards, as demonstration trials in both conditions removed uncertainty about future reward availability [53]. Second, we ruled out the possibility of a lack of numerosity discrimination, considering the ability to differentiate items of various quantities is present from toddlerhood [89, 90]. Finally, the pre-test preference trials (no delay) ensured all children selected the higher quantity. Without a delay, 3-year-olds always opted for the larger reward even from a small difference like 1 versus 2 stickers [91].

Regarding the quantity condition, we adopted that 1:4 ratio. Previous research demonstrated that 4-year-olds showed a strong preference for the later reward using a 1:5 ratio [16]. It is possible that, in our study, children found the quantitative numerical contrast less significant and appealing than the qualitative differences (a glittery animal sticker versus a plain sticker). Notably, the "consequences" of choosing immediate options differed between conditions and it is possible that children considered the plain sticker in the quality condition to be the least favourable, thus were least likely to select it. Additionally, children were able to accumulate rewards through multiple trials. The consecutive gains experienced may downplay their feeling of loss overall, especially in the quantity condition when they received at least one sticker per trial. Therefore, having multiple trials may have left the impression that "*even if I choose the smaller reward this round, I still have something nice, and in the next round I can go for the better one*". We note that our findings were consistent with non-human animal research, which suggests that non-human primates, birds and pigs were better able to delay gratification with rewards differing in quality than quantity [46–51]. Additionally, a similar significant effect of reward type was found in the related study with British children [52].

There was also a main effect of order (Experiment 1) indicating higher performance when they tested on quality-quantity condition than the quantity-quality one. This order effect may be explained in combination with the influence of reward type discussed above. Specifically, children who received the quality condition first may improve performance in later trials. In comparison, the feedback may not be as positive and salient when tested with the quantity condition first, thus it may undermine the subsequent performance in the quality condition. Additionally, with the quantity-quality test order, children received a substantial number of stickers first compared with the quality-quantity order. It is therefore possible that performance suffered more as the attractiveness and novelty of gaining stickers was gradually compromised as the task continued. As a result, motivation (to gain stickers) may decrease, which can influence children's future-oriented decision making [92]. We note that there was no significant order effect in Experiment 2, so perhaps children gained sufficient experience through Experiment 1 (experiencing both conditions) so testing order no longer influenced performance.

We found no influence of reward visibility on performance, which was consistent with Garon et al. [44] delay choice study, yet contrasts with Mischel and Ebbesen's [13] delay maintenance study. These discrepancies may be due to fundamental differences in task designs: the maintenance task requires a continuous presentation of a tempting reward whereas the choice task involves a temporal perspective of immediately obtaining a less-valued reward. Therefore, visual occlusion may aid performance in the maintenance task by reducing the exposure to the arousing reward and decreasing the pre-potency of immediate gratifying responses [13, 14, 54–57]. Additionally, the delay period usually lasted for at least a few minutes [23, 24] whereas in our delay choice study, the delay was only 15 seconds. Furthermore, baiting took place in full view of the children so it was likely that children remembered the locations and did not require external visual cues when making choices. Memory tests were conducted to ensuring children could recall the location of rewards—this was further confirmed in Experiment 2 when both rewards were hidden.

In addition to the rotating tray task, with the Chinese pre-schoolers only, we ran a battery of standardised developmental inhibitory control tasks and one delay choice test. Consistent with existing literature, we found significant age-related performance and correlations between different tests. As a novel finding in our study, there was a significant correlation between performance on the delay choice task [19] and in the quantity condition in Experiment 1 of rotating tray paradigm. Both tests required children to select rewards varying in quantity with no visual barriers, so were most comparable, and the significant results implied convergent validity between these two measures of delayed gratification.

We also found a significant correlation in performance between the Knock-Tap task [82]–a measure of motor inhibition, and the rotating tray task (Experiment 1 & 2). With the latter task, children were required to plan a reaching action to indicate their choices. Thus, in addition to delayed gratification, the rotating tray task also tapped into the ability to control and plan motor movements. The link between self-control and motor domains was not surprising, given their bidirectional interactions highlighted in the dynamic system theory as well as the overlapping brain regions associated with them [93, 94]. We did not find evidence of correlations between the rotating tray task and measures of cognitive inhibition, namely Day-Night task and Grass-Snow task. Two indications can be drawn from these findings. First, this test required suppression of pre-potent responses which does not necessarily entail self-control [2]. Second, the rotating tray task [53], which taps into inhibition and self-control, may share overlapping components with other inhibition tasks, for example the Knock-Tap task, while also being distinctive to different measures of cognitive and motor inhibition. This null finding was consistent with previous literature indicating a weak relationship between executive function inhibitory test and delayed gratification measures [60, for a meta-analysis, see 79].

One notable limitation of the present study was that we only tested the Chinese children with the standardised developmental tasks, thus we are unable to compare British children's performance on rotating tray task with the standardised tasks. This was not due to experimental design; the British dataset was collected first for Miller et al. [52] study and the Chinese dataset was collected later for the present study. Nonetheless, we acknowledge that it would be more comprehensive to have included samples from both countries for all tasks, which could be addressed in future research. Additionally, we highlight a methodological difference between the rotating tray paradigm and the standardised delay choice task [16–19]. The elements of spatial and temporal distance were confounded in the former task as children may select the container based on how close in space and time they were, rather than on reward properties. To address this ambiguity, we administered control and test trials alternating the location of the most preferred rewards, thus ensuring performance reflected ability to decide and act on which rewards were worth waiting for.

It is also important to disentangle the separate cognitive demands implicated in Bramlett et al.'s [53] rotating tray paradigm, as each component may have contributed differently to performance. Therefore, we present a finer analysis of the rotating tray task with reference to children's executive function profile as these are theoretically related constructs [30, 31]. Specifically, working memory was significantly employed in Experiment 2 when reward occlusion was involved. To successfully pass the task, children needed to constantly update information of the spatial locations of the different rewards as control and test trials were administered in a counterbalanced order. Pre-schoolers' performance in working memory has been positively correlated with delay of gratification ability [34–36]. Additionally, previous research has indicated that Chinese preschoolers demonstrate superior working memory to their European-American peers [71, 72, 77].

Thus, it was possible that our Chinese participants had better working memory than the British sample, which contributed to the 4-year-olds and 5-year-olds' superior performance to their British peers. This proposal may explain why we found no significant main effect of country in either experiment, nor any interaction effect with country in Experiment 1 (no visual occlusion) where the demand of working memory was less salient. On a different note, researchers have demonstrated that children with higher inhibition and working memory performed better in tasks of prospective motor control [95]. As aforementioned, motor demands were implicated in the rotating tray task. It is also possible that Chinese children's advantage in working memory and inhibition enabled them to outperform their British counterparts in a task involving motor movements.

In the rotating tray task, the rewards associated with different spatial locations changed per trial. A multiple-trial design placed substantial demands on children's ability to flexibly switch between options and not to preserve their previous selection. Children from Western societies have demonstrated greater inclination to stick with the same rules than Chinese peers [72, 73, 77]. Therefore, it is possible that British pre-schoolers' performance could be undermined by their less-developed cognitive flexibility. Overall, we propose that it was the combined influence from these factors that have led to the outperformance of Chinese 4-year-olds and 5-year-olds over their British counterparts, however we highlight that children performed well overall in both countries. Future work in exploring cross-cultural differences in delayed gratification may include working memory-based tasks with children from both countries and increase the difficulty of the delayed gratification task, such as longer delay lengths. It would also be useful to include measures of parental socialization goals or child-rearing practices [75]. This approach could be incorporated in designing training programs of self-control using qualitatively different rewards and introducing similar strategies into day-to-day parenting.

Finally, we note that, when capuchin monkeys were tested with Bramlett et al.'s [53] rotating tray task, they exhibited delayed gratification even with invisible rewards [59], similar to our findings with children. Due to its intuitive design, the paradigm may potentially impose similar demands on cognitive resources in different species. Comparable findings between species were not surprising if one considers capuchin monkey's performance on cognitive flexibility and working memory. Notably, capuchins monkeys were able to break the cognitive set bias to employ more efficient strategies in an optional switch task [96]. They can attend to and remember different physical stimuli features and retain that information for delay intervals up to 10 minutes [97, 98]. Moreover, capuchin monkeys' forgetting rates were parallel to school-aged children in memory assessment [99]. The rotating tray task therefore offers a valid assessment of delayed gratification across species [53]. More importantly, the comparable results indicate that the ability to delay gratification is biologically and evolutionarily shared across species. Future comparative research could elucidate and compare the influence of contextual factors in other types of self-control tasks across species, such as the reverse contingency and accumulation paradigm.

To summarise, we present the first East versus West comparison of pre-schoolers' delay of gratification using a task originally designed for testing non-human primates [53]. We found critical and replicated age-related performance with older children being more successful at forgoing small immediate rewards for larger valued rewards in the future. Additionally, Chinese 4- and 5-year-olds outperformed their British counterparts, when rewards were occluded. We found correlations in the performance in Chinese children in the rotating tray paradigm with some (though not all tested) standardised developmental inhibitory control and delay choice tasks. By systematically manipulating reward representations, we demonstrated that across Chinese and British children, performance increased when they chose rewards differing in quality over quantity, while visual occlusion had no significant effect. Our study adds to the growing body of cross-cultural research on self-control, and advocates for a diverse and integrated approach to investigate cognitive development in children, with regard to individual differences, as well as at culturally specific levels.

## Supporting information

**S1 Table. Generalized linear mixed models for Experiment 1.** Generalized linear mixed models (final model) on factors affecting the number of correct test and control trials in children. N = Group 1: China 75; Group 2: UK 61. P-values <0.05 are highlighted in bold. The British dataset was published in Miller et al. [52].
(DOCX)

**S2 Table. Generalized linear mixed models for Experiment 2.** Generalized linear mixed models (final model) on factors affecting the number of correct test and control trials in children. N = Group 1: China 75; Group 2: UK 61. P-values <0.05 are highlighted in bold. The British dataset was published in Miller et al. [52].
(DOCX)

**S3 Table. Generalized linear mixed models for Experiment 2 with without outliers.** Generalized linear mixed models (final model) on factors affecting the number of correct *test* trials in children for Experiment 2 with the 3 British 4-year-old outliers removed. N = Britain 58; China 75. P-values <0.05 are highlighted in bold. The British dataset was published in Miller et al. [52].
(DOCX)

**S1 Video.**
(MP4)

## Acknowledgments

Thank you to the staff, parents and children who supported, consented and participated in the study. The British sites were: Histon Early Years Centre, Kennett Community Primary School, Playlanders Playgroup and Preschool, Spinney Primary School, Under Fives Roundabout in Cambridgeshire, and Chearsley & Haddenham Under Fives Preschool and St Mary's CE Combined School in Buckinghamshire. The Chinese site was the affiliated nursery of Yunnan University of Finance and Economics. Special thanks to Ian Millar for assistance with apparatus construction and Ye Ming for advice on video design.

## Author Contributions

**Conceptualization:** Ning Ding, Rachael Miller, Nicola S. Clayton.

**Data curation:** Ning Ding, Rachael Miller.

**Formal analysis:** Ning Ding, Rachael Miller.

**Funding acquisition:** Ning Ding, Nicola S. Clayton.

**Investigation:** Ning Ding, Anna Frohnwieser, Rachael Miller.

**Methodology:** Ning Ding, Rachael Miller, Nicola S. Clayton.

**Project administration:** Ning Ding, Rachael Miller, Nicola S. Clayton.

**Resources:** Ning Ding, Rachael Miller.

**Software:** Rachael Miller.

**Supervision:** Rachael Miller, Nicola S. Clayton.

**Validation:** Ning Ding, Rachael Miller, Nicola S. Clayton.

**Visualization:** Ning Ding.

**Writing – original draft:** Ning Ding.

**Writing – review & editing:** Ning Ding, Anna Frohnwieser, Rachael Miller, Nicola S. Clayton.

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
