## [Decision Letter · Decision Letter 0]

12 Apr 2021

PONE-D-21-06290

Waiting for the Better Reward: Comparison of Delay of Gratification in Young Children across Two Cultures

PLOS ONE

Dear Dr. Miller (Harrison),

Thank you for submitting your manuscript to PLOS ONE. After careful consideration, we feel that it has merit but does not fully meet PLOS ONE’s publication criteria as it currently stands. Therefore, we invite you to submit a revised version of the manuscript that addresses all the points raised by both reviewers during the review process. I feel that both reviewers’ comments were particularly constructive and useful to further strengthen the manuscript.

I particularly agree with Reviewer 1 about the need to further analyze the comparison between Italian and Chinese participants after removing the outliers and with Reviewer 2 on the need to write a more concise Discussion.

I also have some additional comments. Across the manuscript, there are some inconsistencies in the use of the terms delay of gratification and delay choice. As also suggested by Reviewer 1, the paper by Beran (2005) Frontiers in Psychology is an excellent resource to clarify the definitions employed.  

Along with the literature on the stronger role of quality rather than quantity on delay of gratification, please also see: “Addessi, E., & Rossi, S. (2011). Tokens improve capuchin performance in the reverse–reward contingency task. Proceedings of the Royal Society B: Biological Sciences, 278(1707), 849-854.”

Please see some furher, minor comments below:

Introduction, p. 1, second paragraph: not clear what a “standard choice paradigm” is

Introduction, p. 2, second paragraph: not clear why the social perspective is mentioned here, since it is not a topic of the manuscript  

Introduction, p. 3, second paragraph: “her colleagues”

Methods, p. 4, second paragraph: “had to wait”

Discussion, p. 4, second paragraph: “experience” rather than “experiment”?

We look forward to receiving your revised manuscript.

Kind regards,

Elsa Addessi

Academic Editor

PLOS ONE

Journal Requirements:

2. During our internal checks, the in-house editorial staff noted that you conducted research or obtained samples in another country.

Please check the relevant national regulations and laws applying to foreign researchers and state whether you obtained the required permits and approvals.

Please address this in your ethics statement in both the manuscript and submission information.

In addition, please ensure that you have suitably acknowledged the contributions of any local collaborators involved in this work in your authorship list and/or Acknowledgements.

Authorship criteria is based on the International Committee of Medical Journal Editors (ICMJE) Uniform Requirements for Manuscripts Submitted to Biomedical Journals - for further information please see here: https://journals.plos.org/plosone/s/authorship

3. Please remove your figures from within your manuscript file, leaving only the individual TIFF/EPS image files, uploaded separately.  These will be automatically included in the reviewers’ PDF.

4. We note that the Supplementary Material includes footage of participants in the study. 

As per the PLOS ONE policy (http://journals.plos.org/plosone/s/submission-guidelines#loc-human-subjects-research) on papers that include identifying, or potentially identifying, information, the individual(s) or parent(s)/guardian(s) must be informed of the terms of the PLOS open-access (CC-BY) license and provide specific permission for publication of these details under the terms of this license.

Please download the Consent Form for Publication in a PLOS Journal (http://journals.plos.org/plosone/s/file?id=8ce6/plos-consent-form-english.pdf). The signed consent form should not be submitted with the manuscript, but should be securely filed in the individual's case notes.

Please amend the methods section and ethics statement of the manuscript to explicitly state that the patient/participant has provided consent for publication: “The individual in this manuscript has given written informed consent (as outlined in PLOS consent form) to publish these case details”.

If you are unable to obtain consent from the subject of the photograph, you will need to remove the video file and any other textual identifying information or case descriptions for these individuals.

Reviewers' comments:

Reviewer's Responses to Questions

**Comments to the Author**

1. Is the manuscript technically sound, and do the data support the conclusions?

Reviewer #1: Yes

Reviewer #2: Partly

2. Has the statistical analysis been performed appropriately and rigorously? 

Reviewer #1: Yes

Reviewer #2: No

3. Have the authors made all data underlying the findings in their manuscript fully available?

Reviewer #1: No

Reviewer #2: Yes

4. Is the manuscript presented in an intelligible fashion and written in standard English?

Reviewer #1: Yes

Reviewer #2: Yes

5. Review Comments to the Author

Reviewer #1: In the current study (“Waiting for the Better Reward: Comparison of Delay of Gratification in Young Children across Two Cultures”), the authors expanded the research on self-control and delay of gratification with preschoolers to include a more diverse sample of children, including British and Chinese participants. The link with culturally-dependent parenting is very interesting and I appreciate the authors’ efforts in expanding this research. The paper is well-written and should appeal to a broad audience. Because of this (readership by a broad audience), it would be beneficial to expand on some areas of research discussed in the Intro (specific recommendations are listed below).

The research design was straight forward overall, but there are a few outstanding questions concerning inclusion of participants, the inhibition tasks, as well as the final delay of gratification task. It was not clear why Chinese participants were included in the inhibition tasks but not the British. Furthermore, there is a distinct and important difference between self-control tasks (e.g., delay of gratification) and inhibitory tasks (e.g., motor inhibition, Stroop, etc.) – this distinction should be made by the authors as they discuss the correlations between tasks of inhibition and self-control.

• Beran, M. J. (2015). The comparative science of "self-control": What are we talking about? Frontiers in Psychology, 6, Article 51

Can the authors provide a rationale for why they chose the specific inhibitory tasks (Knock Tap, Day Night, Grass Snow) and what the tasks add to the story on developmental self-control? Also, why were these tasks selected (i.e., are they used often in the developmental literature and are they sensitive to age-related changes in inhibition?)

The final dichotomous delay choice task as used (1 sticker now vs. 4 stickers later) has revealed challenging for comparative testing (see papers listed below). The main issue is that a choice for the larger, later option (4 stickers) is synonymous with a prepotent response to select the larger of two rewards. Research with capuchin monkeys has shown that many individuals fail to wait the full amount of time when they are then required to accumulate the chosen rewards (delay maintenance). Thus, the children may have chosen the delayed larger option, which appears like a delayed tolerance of 20 minutes (or the length of the study), but this choice may not accurately reflect children’s self-control or tolerance for such a long delay. The authors also do not include data on children’s performance in the delay choice task except for correlations with the inhibitory tasks (e.g., what proportion of children chose the LL reward? how did performance measure across the 9 trials for participants, etc.?)

• Addessi, E., Paglieri, F., Beran, M. J., Evans, T. A., Macchitella, L., De Petrillo, F., & Focaroli, V.2013). Delay choice versus delay maintenance: Different measures of delayed gratification in capuchin monkeys (Cebus apella). Journal of Comparative Psychology, 127, 392

• Paglieri, F., Focaroli, V., Bramlett, J., Tierno, V., McIntyre, J. M., Addessi, E., ... & Beran, M. J. (2013). The hybrid delay task: Can capuchin monkeys (Cebus apella) sustain a delay after an initial choice to do so?. Behavioural Processes, 94, 45-54.

The authors often reference the rotating tray task as a ‘novel task’ throughout the manuscript, particularly in the Discussion. For example, “novel rotating tray paradigm”, “With a novel delay choice task, we found a…”. The use of ‘novel’ is a bit misleading - please be sure to cite the original work that developed this paradigm, including the original Bramlett paper and follow-ups (see below). Also, the follow ups listed below vary reward visibility for the primates and present the task to new monkeys to test the role of experience in self-control performance – please link the current findings on children with monkeys, particularly as it relates to reward visibility (Perdue et al.)

• Beran, M. J., Perdue, B. M., Rossettie, M. S., James, B. T., Whitham, W., Walker, B., Futch, S. E., & Parrish, A. E. (2016). Self-control assessments of capuchin monkeys with the rotating tray task and the accumulation task. Behavioural Processes, 129, 68-79.

• Perdue, B. M., Bramlett, J. L., Evans, T. A., & Beran, M. J. (2015). Waiting for what comes later: capuchin monkeys show self-control even for nonvisible delayed rewards. Animal cognition, 18(5), 1105-1112.

A lingering question concerns the three outliers in Experiment 2 – as noted by the authors in the Results and Discussion, these three outliers influenced the results of the 4-year-old British group. If the three children that scored zero in test trials for Experiment 2 are removed, does the finding still hold for the country difference? The authors removed these 3 participants to compare 3, 4, and 5-year-olds for the British participants but I am curious if these 3 participants are removed from the group analysis, does the finding still hold that Chinese 4 and 5-year-olds outperform British 4 and 5-year-olds? Please include this information in the Results and if removal of these outliers yields equivalent performance across Chinese and British participants, this should be considered in the Discussion. This seems to be particularly relevant to resolve given that the premise of a country-level (British vs. Chinese) difference potentially lies on these three outliers. The authors argue that Chinese children show greater cognitive flexibility and working memory than the British based on the result that 4 and 5 year old Chinese children outperform 4 and 5 year old British children – if the removal of these three outliers causes this effect to go away, I would suggest that these conclusions be revisited.

More broadly, capuchin monkeys perform quite well in this task over many different manipulations (including the Perdue et al 15 study with nonvisible rewards in the rotating tray) and also show flexibility in responding (they also do not preserve their former choice but perform well even for non-visible rewards and when placed in different locations), and so I wonder if the authors would consider the monkeys’ working memory and cognitive flexibility to be on par with the children? Please incorporate some discussion of the comparison of capuchin monkey and children performance across the different tasks/conditions using the rotating tray and what the similarity in responding across species may indicate.

I also provide comments below for each section as well a Minor Comments at the end.

Introduction:

Second Paragraph (“Parallel to…”) - Can the authors expand some on what we know about children’s performance in the delay maintenance and delay choice tasks? What types of rewards (primary or secondary) are typically used for children in these studies, what delay lengths (e.g., seconds or minutes, etc.) are typically shown by 3-year-olds vs. 4-year-olds (is there a large difference in tolerance for delay across this age-span? How do preschoolers compare in these tasks to older children?)

Fourth Paragraph (“Theoretically…”) – Can the authors expand on the references 25-26 (i.e., were inhibitory control and delay of gratification measured in these tasks, how, please expand?). Can the authors give more detail for Reference 30 (i.e., what type of intervention was used? how long were the benefits of training seen? etc.).

Research with nonhuman primates (and pigs) reveals a similar effect of quality > quantity as described for avian species – I would recommend including the following articles when referencing comparative work on these variables in the Introduction & Discussion:

• Anderson, J. R., Hattori, Y., & Fujita, K. (2008). Quality before quantity: Rapid learning of reverse-reward contingency by capuchin monkeys (Cebus apella). Journal of Comparative Psychology, 122(4), 445–448.

• Beran, M. J., & Evans, T. A. (2006). Maintenance of delay of gratification by four chimpanzees (Pan troglodytes): The effects of delayed reward visibility, experimenter presence, and extended delay intervals. Behavioural processes, 73(3), 315-324.

• Glady, Y., Genty, É., & Roeder, J. J. (2012). Brown Lemurs (Eulemur fulvus) can master the qualitative version of the reverse-reward contingency. Plos one, 7(10), e48378.

• Zebunke, M., Kreiser, M., Melzer, N., Langbein, J., & Puppe, B. (2018). Better, not just more—contrast in qualitative aspects of reward facilitates impulse control in pigs. Frontiers in psychology, 9, 2099.

Methods/Results:

Day-Night and Grass-Snow tasks – was response time measured in addition to accuracy? RT data is considered more sensitive than accuracy data and thus more informative regarding inhibition ‘failures’ for standard Stroop tasks with adults.

Table 1 can be removed and incorporated into the text – the average age of all participants for each country is sufficient (with standard deviation) versus the average for each year

Testing Q – were children immediately given their selected reward to keep? Include this in the Testing description.

Figure 2 caption needs more information. Perhaps label the boxes as 1 and 2 and provide some information about the symbols in the caption.

Because there was an age effect in Experiment 1, it may be helpful to show this in Figure 3 as the authors depicted in Figure 4. Right now, Figure 3 focuses on the country difference (which was a null finding) and is collapsed across age (which is the significant effect).

See above for question regarding removal of the three outliers for the British vs. Chinese age comparison.

Results were missing for the final Delay of Gratification task (smaller, sooner vs. larger, later) – what proportion of children chose the LL reward? how did performance measure across the 9 trials for participants, etc.?

Discussion

In the first paragraph, be sure to specify that British children were not presented with the other inhibition tasks as it currently reads as if this effect was only found for Chinese children.

When presenting the rotating tray task as “novel” throughout the Discussion, the authors should cite the original work using this paradigm as this task. For example, instead of “With a novel delay choice task, we found a…” -- “Using a test designed for comparative assessments of self-control in primates (Bramlett et al.), we found a…”

The authors focus heavily on potential differences in working memory and cognitive flexibility of Chinese vs. British, but see above for question regarding three outliers. Also, include some discussion of capuchin monkey vs. children performance as it pertains to cognitive flexibility and working memory (including the Perdue et al 15 study with capuchins and nonvisible rewards).

In the last sentence, include information on age-related changes as well as this is a critical and replicated result.

Minor Comments:

Introduction:

It is odd to switch from “we” to “they” in the 2nd sentence of the Introduction

Change “on” to “in” for this sentence “in the face of” is the common wording for this phrase – “The measure of interest is the length of time lapsed as children need to maintain their action IN the face of…”

Change from “posting to “posing” in the following sentence - “Furthermore, the role of reward representation has not been examined in East Asian cultures, POSING important questions…”

Change “varying” to “varied” – “we expected to see similar patterns to those in Miller et al. (42) and Garon et al. (37) where children performed better when reward VARIED in quality…”

Methods/Results:

Remove the word “with” – “In Britain, we recruited and tested 61 children at schools in Cambridgeshire and Buckinghamshire, which served predominantly white…”

Change the word “to” to “in” – “The Chinese data was collected specifically for the present study and this data set has not been used IN any other publications to date.”

Change “experimenter” to plural – “all EXPERIMENTERS followed a prepared...”

Change “than” to “then” – “half of children received the quantity condition THEN the quality condition…”

Change “significant” to “significantly” –“Moreover, 4-year-olds scored SIGNIFICANTLY higher on the Knock-Tap, Day-Night and Grass-Snow task than children aged 3-years-old (all p <. 02).”

Change “task” to “tasks” – “Additionally, we found that children’s performance in the quantity condition in Experiment 1 in the rotating tray paradigm was significantly correlated with performance in the standard delay of gratification task, both TASKS involving choices...”

Discussion:

Change “chimpanzee” to plural – “We note that our findings are consistent with contemporary research on non-human animals, which suggests that corvids, parrots and CHIMPANZEES are able…”

Change “tend’ to “trend” – “This proposal may also explain why we only found non-significant TREND effect of country in...”

Change “emphasis” to “emphasize” – “We do EMPHASIZE again, however, that both groups performed well overall.”

Change “Out” to “Our” – “OUR study not only add depth to the growing body of cross-cultural research on self-control…”

Reviewer #2: The authors report an experiment on Chinese preschooler’s delay of gratification, using a novel apparatus/procedure that had previously been used in comparative studies. They also included a series of inhibitory control measures and manipulated reward visibility and quantity/quality. Data were compared with data from British preschoolers from a previous study. The studies were well conducted, the manuscript is detailed and well-structured, and the findings are interesting.

I have some comments and suggestions for the authors:

(1) Introduction: The introduction gives a good overview of the state of the field and motivates the purpose of the study well.

(1a) However, it took me until the methods section to realize that only the Chinese data were novel data and that the British data were based on data from a previous study. This also explained my puzzlement about the fact that additional inhibitory control measures were collected for the Chinese sample, but not for the British sample. I think the authors could centre their introduction more on the novel Chinese data they collected and explicitly state that British data from a previous study was used for comparison. I think it would also be helpful to clarify this point in the abstract (i.e., distinguish between novel and previously collected data).

(1b) There were some instances where expressions were non-idiomatic or sounded somewhat awkward (“..need to maintain their action on the face of a tempting treat…”, “children of 4 years-old and above”, “..Chinese show advantageous performance on self-control tasks than their..”, etc.). There were further instances throughout the manuscript, and I think the manuscript would benefit from careful proof-reading and language editing.

(2) Methods: The methods are clearly described.

(2a) Participants: Further details on Chinese participants/data collection would be helpful. For example, when was the Chinese data collected? How many Chinese children per age group were tested for the study (currently, only numbers for British and Chinese children together are provided)?

(2b) The authors describe their counterbalancing within Experiments; was the order of Experiments also counterbalanced or did they take place in a fixed order?

(2c) Delay of gratification task: The authors stated that the delayed reward remained inaccessible until the end of the study. How long was the delay and was the delay standardized across children (or did it vary)?

(2d) Interrater reliability: usually Cohen’s Kappa or interrater-reliability coefficients (ICC) are reported as measures of reliability and not just percentage agreement. It would be helpful to add one of these standard measures.

(3) Analyses: Using GLMMs to analyse the data is an adequate choice and full-null model comparisons are helpful to ensure the variables of interest predict outcome measures.

(3a) It would be good if the authors could add further details on their analytical approach: for example, which control variables did the null models contain? What function did they use for likelihood ratio tests to determine p-values of model predictors? For transparency, it would be helpful if the authors published their R code alongside their paper or, as a minimum, provide details on their model-formulas in the manuscript.

(3b) Furthermore, the authors may consider comparing a full model (with interaction term) with a reduced model (only main effects) to find the model with the best fit to the data, and then report the best fit model and its results.

(3c) What really puzzled me (and, in fact, seemed redundant) are the non-parametric follow-up tests. The GLMMs already provide all the relevant information (i.e., is there a significant effect of country, condition, age, etc.). There are also packages available in R for follow-up tests for predictors with more than 2 levels (e.g., the lsmeans package). Also, while age is entered as continuous variable in the models, it is then used as a categorical variable in the follow up tests. If the authors were interested in differences between age groups, they could have entered age as categorical variable in their GLMMs.

(4) Results:

(4a) The studies also included preference and control trials, but the data are currently not reported. I think it would be helpful to report these data in the results section.

(4b) Table 2: it would be helpful to clarify that the p-values were derived from likelihood ratio tests.

(4c) Please also see previous comments that it is not clear why non-parametric tests are reported.

(5) The discussion elaborates on the findings in detail. It could have been more concisely written at times, but I leave it up to the authors whether they would want to edit the discussion section.

Minor comment:

1st paragraph, introduction: “From financial decision in humans to foraging behaviours in other animals, we frequently face intertemporal choices in which they weigh the costs and and benefits…” This sentence sounds as if “we” are also “other animals” and switches somewhat awkwardly between “we” and “they”. Reword?

For future submissions, it would be helpful if the manuscript file included page numbers, so they can be used as reference for comments.

6. PLOS authors have the option to publish the peer review history of their article (what does this mean?). If published, this will include your full peer review and any attached files.

Reviewer #1: No

Reviewer #2: No

---

## [Author Response · Author response to Decision Letter 0]

31 May 2021

Dear Dr Addessi,

Following an invitation to revise [PONE-D-21-06290], we would like to submit our manuscript entitled “Waiting for the Better Reward: Comparison of Delay of Gratification in Young Children across Two Cultures”. 

We wish to thank you and the two reviewers for the helpful and constructive comments. We have now fully revised the manuscript and accompanying documents in accordance with these comments. Please find responses to each comment in this response to reviewers’ document. Please note that line numbers correspond with the tracked changes version of the manuscript.

We hope that following our revisions, you will consider our manuscript for publication in PlOS One.

Yours Sincerely.

Ning Ding, Anna Frohnwieser, Rachael Miller, Nicky Clayton

Dear Dr. Miller (Harrison),

Thank you for submitting your manuscript to PLOS ONE. After careful consideration, we feel that it has merit but does not fully meet PLOS ONE’s publication criteria as it currently stands. Therefore, we invite you to submit a revised version of the manuscript that addresses all the points raised by both reviewers during the review process. I feel that both reviewers’ comments were particularly constructive and useful to further strengthen the manuscript.

I particularly agree with Reviewer 1 about the need to further analyze the comparison between Italian and Chinese participants after removing the outliers and with Reviewer 2 on the need to write a more concise Discussion.

We greatly appreciated the Editor and Reviewers’ constructive feedback regarding the influence of outliers. Please see our detailed responses to Reviewer 1’s Question 8 (Page 6) regarding this point.

I also have some additional comments. Across the manuscript, there are some inconsistencies in the use of the terms delay of gratification and delay choice. As also suggested by Reviewer 1, the paper by Beran (2005) Frontiers in Psychology is an excellent resource to clarify the definitions employed.

Thanks for the very useful reference. We have ensured the use of terminology is consistent throughout the manuscript and we also included new sections on the distinctions between self-control and inhibition (line 117-123).

Along with the literature on the stronger role of quality rather than quantity on delay of gratification, please also see: “Addessi, E., & Rossi, S. (2011). Tokens improve capuchin performance in the reverse–reward contingency task. Proceedings of the Royal Society B: Biological Sciences, 278(1707), 849-854.”

Thanks for the recommendation for further reference. We have included Addessi & Rossi (2011) when introducing and discussing the role of reward type (line 154-155)

Please see some further, minor comments below:

Introduction, p. 1, second paragraph: not clear what a “standard choice paradigm” is

We have worked on the consistency of terminology of different tasks and ensured there is no ambiguity when referring to specific tests.

Introduction, p. 2, second paragraph: not clear why the social perspective is mentioned here, since it is not a topic of the manuscript.

We included these additional studies to demonstrate that a wide range of contextual factors can influence children’s performance on delay of gratification tasks. We feel that the content is sensible and informative to be included briefly for the purpose of introduction. 

Introduction, p. 3, second paragraph: “her colleagues”

We have changed the format of citation in the sentence (line 175).

Methods, p. 4, second paragraph: “had to wait”

Changed to “need to wait” (line 360).

Discussion, p. 4, second paragraph: “experience” rather than “experiment”?

Changed to “experience” (line 716).

Reviewer #1: In the current study (“Waiting for the Better Reward: Comparison of Delay of Gratification in Young Children across Two Cultures”), the authors expanded the research on self-control and delay of gratification with pre-schoolers to include a more diverse sample of children, including British and Chinese participants. The link with culturally-dependent parenting is very interesting and I appreciate the authors’ efforts in expanding this research. The paper is well-written and should appeal to a broad audience. Because of this (readership by a broad audience), it would be beneficial to expand on some areas of research discussed in the Intro (specific recommendations are listed below).

Thank you very much for your constructive feedback. 

The research design was straight forward overall, but there are a few outstanding questions concerning 

1). Inclusion of participants, the inhibition tasks, as well as the final delay of gratification task. 

Inclusion of participants: At the broader level, the main aim of the study was to investigate Eastern-Western contrast on children’s performance of delay of gratification using the rotating tray developed by Bramlett et al. (2012). The reasoning behind the inclusion of Chinese preschoolers were that as a typical example of Eastern culture and society, the Chinese populations has been the focal point of many cross-cultural developmental research (Lan et al., 2011; Sabbagh et al., 2006; Xu et al., 2020). Therefore, with Miller et al. (2019) data on British children, we feel that the inclusion of a Chinese dataset would allow us to first apply Bramlett et al. (2012) rotating tray paradigm to a different cultural group and assess its validity, and second to compare children’s self-control development from different cultural contexts. We have provided a brief justification on the inclusion of Chinese participants (line 213-215).

Inclusion of the inhibition tasks: Theoretically, not all tasks require self-control involves behavioural inhibition (Beran, 2015). Yet, as Bramlett et al. (2012) described, the rotating tray task involves the ability to make and act-upon future-oriented decision as well as the inhibition of reaching-and-taking responses for the sooner reward. Therefore, there are potentially overlapping components between this specific self-control task and measures of inhibition. To date there has been no investigation dedicated to clarifying the underlying relationship between the comparative task of delay of gratification and inhibitory control tests among pre-schoolers. Therefore, we aimed to address this literature gap and we feel our results would shed light on our understanding of the similarities and differences among various self-control and behavioural inhibition tests. We have briefly explained our rationale of including the inhibitory control tasks in the introduction (line 229-235).

Inclusion of the standard delay of gratification task: Theoretically, the use of different measures on the same construct would inform about the convergent validity of different tests and enable researchers to validate and compare tasks. Few studies have explored whether delay of gratification performance is consistent within individuals when tested using different paradigms (Miller et al., 2019). As Bramlett proposed in the original paper (Bramlett et al., 2012), data from the rotating tray task should be compared to other self-control tests. Given these considerations, we therefore decided to adopt a battery of tasks on delayed gratification, specifically, the more recent paradigm of rotating tray task (Bramlett et al., 2012) and a widely used delay choice task (Prencipe & Zelazo, 2005). The delay choice task was selected over delay maintenance task because (1) the structure of the delay choice task was more similar to the rotating tray paradigm, and (2) concerns of study length (delay maintenance paradigm can take up to 20 mins. We have added more information on the inclusion of standardised delay choice task (line 450-452).

2). It was not clear why Chinese participants were included in the inhibition tasks but not the British.

The British data came from Miller et al. (2019), a comparative study focusing on children and New Caledonian crows, and this cross-species study was designed to test for delay of gratification ability using only the Bramlett et al. (2012) paradigm. Therefore, it did not include any standardised child developmental measures of inhibitory control. The current study was conducted after the completion of data collection of Miller et al. (2019). Due to the difficulty of tracking previous participants, obtaining parental consent and the fact that the previously tested children were now older, it was not feasible for us to test the same group of British children on the battery of inhibitory tasks that we used with the Chinese children. Moreover, we treated the current study as an independent cross-cultural investigation of children’s delay of gratification with its own rationale and methodology. We’ve recruited adequate number of participants in China to ensure enough power when analyzing children’s performance on the inhibitory control tasks. We have elaborated on why British children were not included in the inhibition and standardized delay choice task (line 288-293). 

3). Furthermore, there is a distinct and important difference between self-control tasks (e.g., delay of gratification) and inhibitory tasks (e.g., motor inhibition, Stroop, etc.) – this distinction should be made by the authors as they discuss the correlations between tasks of inhibition and self-control.

• Beran, M. J. (2015). The comparative science of "self-control": What are we talking about? Frontiers in Psychology, 6, Article 51

We appreciate the reviewer’s suggestion on this issue and we agree that it is extremely relevant and important in the current paper. We have added new sections in the introduction and discussion regarding the distinctions between self-control tasks and inhibitory control tasks (line 117-123, line 747-753). 

4). Can the authors provide a rationale for why they chose the specific inhibitory tasks (Knock Tap, Day Night, Grass Snow) and what the tasks add to the story on developmental self-control? Also, why were these tasks selected (i.e., are they used often in the developmental literature and are they sensitive to age-related changes in inhibition?)

Day-Night, Grass-Snow and Knock-Tap tasks are all members of a family referred as Stimulus-Response Compatibility tasks (Simpson, Upson, & Carroll, 2017). A recent meta-analytic study has revealed that along with Go/No-Go tasks, they were the two most widely used inhibitory control tests in developmental literature (Petersen et al., 2016). In addition to being valid and reliable measures of inhibitory control, they have shown to be sensitive to age-related changes and researchers have repeatedly documented significant developmental trajectories using these tasks (Carlson, 2005). In the current study, we aimed to capture the different types of inhibitory control, i.e. verbal inhibition (Day-Night) and motor inhibition (Knock-Tap). The Grass-Snow task involves highly pre-potent declarative pointing and it was selected as a variant version of the original Day-Night task. We have provided a brief justification on why these inhibitory control tasks were selected (line 418-419). 

Regarding the question of what the tasks add to the study on developmental self-control, given the potential overlapping component of behavioural inhibition in rotating tray paradigm and the standardised developmental inhibitory control tasks in humans, we feel the knowledge of how Bramlett et al (2012) delay choice paradigm correlates with standardised inhibitory control tasks sheds light on our understanding of the similarities and differences among various self-control and behavioural inhibition tests (line 231-235). 

5). The final dichotomous delay choice task as used (1 sticker now vs. 4 stickers later) has revealed challenging for comparative testing (see papers listed below). The main issue is that a choice for the larger, later option (4 stickers) is synonymous with a prepotent response to select the larger of two rewards. Research with capuchin monkeys has shown that many individuals fail to wait the full amount of time when they are then required to accumulate the chosen rewards (delay maintenance). Thus, the children may have chosen the delayed larger option, which appears like a delayed tolerance of 20 minutes (or the length of the study), but this choice may not accurately reflect children’s self-control or tolerance for such a long delay. The authors also do not include data on children’s performance in the delay choice task except for correlations with the inhibitory tasks (e.g., what proportion of children chose the LL reward? how did performance measure across the 9 trials for participants, etc.?)

• Addessi, E., Paglieri, F., Beran, M. J., Evans, T. A., Macchitella, L., De Petrillo, F., & Focaroli, V.2013). Delay choice versus delay maintenance: Different measures of delayed gratification in capuchin monkeys (Cebus apella). Journal of Comparative Psychology, 127, 392

• Paglieri, F., Focaroli, V., Bramlett, J., Tierno, V., McIntyre, J. M., Addessi, E., ... & Beran, M. J. (2013). The hybrid delay task: Cancapuchin monkeys (Cebus apella) sustain a delay after an initial choice to do so?. Behavioural Processes, 94, 45-54.

We agree that there is a methodological issue in the dichotomous delay choice task (Prencipe & Zelano, 2005) that the larger delayed option is synonymous with prepotent responses to select the larger of two rewards. Indeed, this was the issue that Bramlett et al. (2012) aimed to address with the rotating tray paradigm. Despite its limitations, the delay choice task has been widely used in developmental literature as a standandarised measure of delay of gratification. We presented the results of the delay choice task in Table 3 which shown age significantly influenced children’s performance across countries and also post-hoc comparisons between different age groups (Table 3, line 607-612). We did not conduct trial by trial analysis for the delay choice task but focused on the sum score of correct trials. The reasoning behind the scoring was that this method is typical when investigating children age-related changes of delayed gratification ability and has been validated in numerous prior research (Imuta et al., 2015; Lemmon & Moore, 2007; Thompson et al., 1997). Additionally, we included the standardised delay choice task to correlate children’s performance with the rotating tray task, which would use the sum score for analysis. Therefore, for the scope of the current study, we feel that our choice of task scoring was well supported by developmental literature and we have presented sufficient information regarding children’s performance on the standardised delay choice task. We have added a brief justification of the scoring method in the methods section (line 500-504).

6). The authors often reference the rotating tray task as a ‘novel task’ throughout the manuscript, particularly in the Discussion. For example, “novel rotating tray paradigm”, “With a novel delay choice task, we found a…”. The use of ‘novel’ is a bit misleading - please be sure to cite the original work that developed this paradigm, including

the original Bramlett paper and follow-ups (see below). 

The terminology used to describe the rotating tray task has been made consistent throughout the manuscript with reference to the original Bramlett et al. (2012) paper.

7). Also, the follow ups listed below vary reward visibility for the primates and present the task to new monkeys to test the role of experience in self-control performance – please link the current findings on children with monkeys, particularly as it relates to reward visibility (Perdue et al.)

• Beran, M. J., Perdue, B. M., Rossettie, M. S., James, B. T.,Whitham, W., Walker, B., Futch, S. E., & Parrish, A. E. (2016). Self-control assessments of capuchin monkeys with the rotating tray task and the accumulation task. Behavioural Processes, 129, 68-79.

• Perdue, B. M., Bramlett, J. L., Evans, T. A., & Beran, M. J. (2015). Waiting for what comes later: capuchin monkeys show self-control even for nonvisible delayed rewards. Animal cognition, 18(5), 1105-1112. 

We thank the reviewer for these reference recommendations. We have ensured the relevant studies were included in the introduction and discussion on the section of reward visibility. 

8). A lingering question concerns the three outliers in Experiment 2 – as noted by the authors in the Results and Discussion, these three outliers influenced the results of the 4-year-old British group. If the three children that scored zero in test trials for Experiment 2

are removed, does the finding still hold for the country difference? The authors removed these 3 participants to compare 3, 4, and 5-year-olds for the British participants but I am curious if these 3 participants are removed from the group analysis, does the finding

still hold that Chinese 4 and 5-year-olds outperform British 4 and5-year-olds? Please include this information in the Results and if removal of these outliers yields equivalent performance across Chinese and British participants, this should be considered in the Discussion. 

We have taken two steps to tackle the issue regarding the three potential outliers of British 4-year-olds. First, we have re-run the GLMM analysis for models with the Experiment 2 dataset without the 3 British 4-year old outliers and the fixed term “country” still has as significant effect. Therefore, the three outliers did not influence the overall GLMM results. We have added the table (S3 Table) to the section of supporting information and referred to it in the results section in the manuscript (i.e. that removing the 3 outliers produces the same overall results regarding the significant effects) (line 595-603). Secondly, we have re-run the post-hoc test to compare Chinese 4 vs British 4-year-olds after the removal of the three outliers and importantly there is a still significant country difference (Z = 3.992, P <.001). The results of Chinese and British 5-year-olds remained the same since there was no outliers to be removed from the analysis (Z = 3.148, P = .002) (supporting information). 

The additional analysis described above indicates that the three outliers of British 4-year-olds did not cause any significant distortions to our results regarding children’s overall age-related performance and country effects on the rotating tray task, or influence the specific comparisons between British 4-year-old and Chinese 4-year-olds. Therefore, we feel these additional analyses are sufficient to clarify the outliers did not influence our results and provide firm support of the country effect found between Chinese and British pre-schoolers. We have decided to report the results based on the full dataset in the manuscript and include selected result outputs (GLMM and post-hoc comparison between 4-year-olds) based on the dataset without outliers in the supporting information (S3 Table). 

The justification for this are that, before data collection we did not specify any criteria for removal of potential outliers, and removing the outliers completely would not be a full and comprehensive representation of the data we collected. Second, given there is no significant changes to our results in removing these 3 outliers, particularly the significant contrast between British and Chinese 4-year-olds, we present the full dataset to the readers while acknowledging the presence of these three outliers in the discussion (line 665-667). The information included in the supporting information will allow readers to have access to results derived from both datasets and we feel this solution should be both informative and scientifically rigorous. 

9). This seems to be particularly relevant to resolve given that the premise of a country-level (British vs. Chinese) difference potentially lies on these three outliers. The authors argue that Chinese children show greater cognitive flexibility and working memory than the British based on the result that 4- and 5-year-old Chinese children outperform 4- and 5-year-old British children – if the removal of these three outliers causes this effect to go away, I would suggest that these conclusions be revisited.

Please see our responses to Question 8 regarding the potential influence of these three outliers. The additional analysis indicates that there is still a significant country level difference between Britain and China among the 4-year-olds when the outliers were removed. Moreover, the discussion on working memory and cognitive flexibility was closely related to the cognitive demands involved in the rotating tray task. Therefore, we feel our discussion for the cross-cultural differences remains reasonable and valid. 

(10). More broadly, capuchin monkeys perform quite well in this task over many different manipulations (including the Perdue et al 15 study with nonvisible rewards in the rotating tray) and also show flexibility in responding (they also do not preserve their former choice but perform well even for non-visible rewards and when placed in different locations), and so I wonder if the authors would consider the monkeys’ working memory and cognitive flexibility to be on par with the children? Please incorporate some discussion of the comparison of capuchin monkey and children performance across the different

tasks/conditions using the rotating tray and what the similarity in responding across species may indicate.

We have added research on capuchin monkey’s working memory and cognitive flexibility and inserted new sections to discuss the comparable results between non-human animals and human children (line 801-815).

I also provide comments below for each section as well a Minor Comments at the end.

Introduction:

11). Second Paragraph (“Parallel to…”) - Can the authors expand some on what we know about children’s performance in the delay maintenance and delay choice tasks? What types of rewards (primary or secondary) are typically used for children in these studies, what delay lengths (e.g., seconds or minutes, etc.) are typically shown by 3-year-olds

vs. 4-year-olds (is there a large difference in tolerance for delay across this age-span? How do preschoolers compare in these tasks to older children?)

We have expanded the content of children’s performance in the delay maintenance task and delay choice task and ensured that it covered the points raised by the reviewer (line 92-106). 

12). Fourth Paragraph (“Theoretically…”) – Can the authors expand on the references 25-26 (i.e., were inhibitory control and delay of gratification measured in these tasks, how, please expand?). 

We have elaborated on reference 25-26 (now reference 34-35) and provided more details (line 130-132).

13). Can the authors give more detail for Reference 30 (i.e., what type of intervention was used? how long were the benefits of training seen? etc.). 

We have included more information about the intervention study (line 135-138). 

14).Research with nonhuman primates (and pigs) reveals a similar effect of quality > quantity as described for avian species – I would recommend including the following articles when referencing comparative work on these variables in the Introduction & Discussion:

• Anderson, J. R., Hattori, Y., & Fujita, K. (2008). Quality before quantity: Rapid learning of reverse-reward contingency by capuchin monkeys (Cebus apella). Journal of Comparative Psychology, 122(4), 445–448.

• Beran, M. J., & Evans, T. A. (2006). Maintenance of delay of gratification by four chimpanzees (Pan troglodytes): The effects of delayed reward visibility, experimenter presence, and extended delay intervals. Behavioural processes, 73(3), 315-324.

• Glady, Y., Genty, É., & Roeder, J. J. (2012). Brown Lemurs (Eulemur fulvus) can master the qualitative version of the reverse-reward contingency. Plos one, 7(10), e48378.

• Zebunke, M., Kreiser, M., Melzer, N., Langbein, J., & Puppe, B. (2018). Better, not just more—contrast in qualitative aspects of reward facilitates impulse control in pigs. Frontiers in psychology, 9, 2099.

We greatly appreciated the reviewer’s suggestion of additional references. We have ensured the listed studies of comparative work were included in introduction and discussion (line 153-154, line 702-704). 

Methods/Results:

15). Day-Night and Grass-Snow tasks – was response time measured in addition to accuracy? RT data is considered more sensitive than accuracy data and thus more informative regarding inhibition ‘failures’ for standard Stroop tasks with adults.

We agree that response time, especially with computerized tasks, provides more comprehensive data than simply focusing on accuracy in adult literature. In developmental psychology, accuracy score has been adopted across a wide range of tasks, for example, inhibitory control and cognitive flexibility. The tasks we used in the manuscript (Day-Night, Grass-Snow, and Knock-Tap) have all proved to be developmentally sensitive measures for pre-schoolers (Carlson, 2005, Petersen et al., 2016) and research using same experimental paradigms have dominantly focused on accuracy scores. Therefore, we feel that our scoring and analysis of these tasks are reliable and consistent with existing literature. We have provided a brief justification on the scoring method of the inhibitory control tasks (line 500-501). 

16). Table 1 can be removed and incorporated into the text – the average age of all participants for each country is sufficient (with standard deviation) versus the average for each year. 

We have removed table 1 and incorporated the participants’ demographics into the main text (line 273-283). We used separate mean and range to illustrate the participant demographics in each country. The reasons for using range instead of standard deviation are: 1). range provides a more comprehensive overview of the youngest and oldest children in each age group; 2) to be consistent with Miller et al. (2019) where the same group of British children were tested and the authors used mean and range in the methods section. 

17). Testing Q – were children immediately given their selected reward to keep? Include this in the Testing description.

Across Experiment 1 and 2, children would be given their selected rewards immediately after the disk stopped rotation (line 369). 

18). Figure 2 caption needs more information. Perhaps label the boxes as 1 and 2 and provide some information about the symbols in the caption. 

We had changed the captions and symbols in Fig 2 as suggested (line 400-402). 

Because there was an age effect in Experiment 1, it may be helpful to show this in Figure 3 as the authors depicted in Figure 4. Right now, Figure 3 focuses on the country difference (which was a null finding) and is collapsed across age (which is the significant effect). 

We have prepared a figure for the significant age-effect in Experiment 2 (Fig 5) and deleted the graph on the null finding of country effect as suggested. 

19). See above for question regarding removal of the three outliers for the British vs. Chinese age comparison.

Please see our above responses to Question 9 and 10 regarding the removal of outliers. 

20). Results were missing for the final Delay of Gratification task (smaller, sooner vs. larger, later) – what proportion of children chose the LL reward? how did performance measure across the 9 trials for participants, etc.?

Please see our responses to Question 5 regarding the scoring and results for the standardized delay choice task. 

Discussion

21). In the first paragraph, be sure to specify that British children were not presented with the other inhibition tasks as it currently reads as if this effect was only found for Chinese children.

We have explicitly stated that the correlational findings only applied to Chinese children and we did not test inhibitory control with British children (line 645-648). 

22). When presenting the rotating tray task as “novel” throughout the Discussion, the authors should cite the original work using this paradigm as this task. For example, instead of “With a novel delay choice task, we found a…” -- “Using a test designed for comparative assessments of self-control in primates (Bramlett et al.), we found

a…” 

We have ensured the use of terminology is consistent to describe the rotating tray task with reference to the original Bramlett et al. (2012) paper.

23). The authors focus heavily on potential differences in working memory and cognitive flexibility of Chinese vs. British but see above for question regarding three outliers. 

Please see our above responses to Question 9 and Question 10 regarding the issue of three outliers. Given the cross-cultural difference remains significant after the removal of the three outliers of British 4-year-olds, we feel our discussion of working memory and cognitive flexibility is valid and sensible in explaining the country differences. 

24). Also, include some discussion of capuchin monkey vs. children performance as it pertains to cognitive flexibility and working memory (including the Perdue et al 15 study

with capuchins and nonvisible rewards).

We have included a separate paragraph to discuss the comparable performance between capuchin monkeys and children (line 801-815).

25). In the last sentence, include information on age-related changes as well as this is a critical and replicated result.

We have stated in the last paragraph that the age-related performance is a critical and replicated result in preschool years (line 817-819). 

Minor Comments:

Introduction:

26). It is odd to switch from “we” to “they” in the 2nd sentence of the Introduction 

We have removed “we” and reworded the sentence to avoid any confusion (line 74). 

27). Change “on” to “in” for this sentence “in the face of” is the common wording for this phrase – “The measure of interest is the length of time lapsed as children need to maintain their action IN the face of…”

We have changed “on” to “in” (line 89). 

28). Change from “posting to “posing” in the following sentence - “Furthermore, the role of reward representation has not been examined in East Asian cultures, POSING important questions…” 

We have corrected the typo (line 210).

29). Change “varying” to “varied” – “we expected to see similar patterns to those in Miller et al. (42) and Garon et al. (37) where children performed better when reward VARIED in quality…”

We have changed the word to “varied” as suggested (line 252). 

Methods/Results:

30). Remove the word “with” – “In Britain, we recruited and tested 61 children at schools in Cambridgeshire and Buckinghamshire, which served predominantly white…”

“With” has been removed.

31). Change the word “to” to “in” – “The Chinese data was collected specifically for the present study and this data set has not been used IN any other publications to date.”

“to” has been changed to “in” (line 288). 

32). Change “experimenter” to plural – “all EXPERIMENTERS followed a prepared...”

We have changed the wording in the sentence (line 340). 

33). Change “than” to “then” – “half of children received the quantity condition THEN the quality condition…

The typo has been corrected (line 412). 

34). Change “significant” to “significantly” –“Moreover, 4-year-olds scored SIGNIFICANTLY higher on the Knock-Tap, Day-Night and Grass-Snow task than children aged 3-years-old (all p <. 02).” 

We have made the change from “significant” to “significantly” (line 610).

35). Change “task” to “tasks” – “Additionally, we found that children’s performance in the quantity condition in Experiment 1 in the rotating tray paradigm was significantly correlated with performance in the standard delay of gratification task, both TASKS involving choices...”

We have used the plural “tasks” in the sentence (line 630). 

Discussion:

36). Change “chimpanzee” to plural – “We note that our findings are consistent with contemporary research on non-human animals, which suggests that corvids, parrots and CHIMPANZEES are able…”

We have used different words in the sentences, “non-human primates and birds” (line 702).

37). Change “tend’ to “trend” – “This proposal may also explain why we only found non-significant TREND effect of country in...”

We have deleted the sentence. 

38). Change “emphasis” to “emphasize” – “We do EMPHASIZE again, however, that both groups performed well overall.”

We have changed the word to “highlight” (line 794). 

38). Change “Out” to “Our” – “OUR study not only add depth to the growing body of cross-cultural research on self-control…”

The typo has been corrected (line 826). 

Reviewer #2: The authors report an experiment on Chinese preschooler’s delay of gratification, using a novel apparatus/procedure that had previously been used in comparative studies. They also included a series of inhibitory control measures and manipulated reward visibility and quantity/quality. Data were compared with data from

British preschoolers from a previous study. The studies were well conducted, the manuscript is detailed and well-structured, and the findings are interesting. I have some comments and suggestions for the authors:

Thank you for your helpful advice and comments. 

(1) Introduction: The introduction gives a good overview of the state of the field and motivates the purpose of the study well.

(1a) However, it took me until the methods section to realize that only the Chinese data were novel data and that the British data were based on data from a previous study. This also explained my puzzlement about the fact that additional inhibitory control measures were collected for the Chinese sample, but not for the British sample. 

I think the authors could centre their introduction more on the novel Chinese data they collected and explicitly state that British data from a previous study was used for comparison. I think it would also be helpful to clarify this point in the abstract (i.e., distinguish between novel and previously collected data).

We appreciate the reviewer’s suggestions. We have now explicitly stated in the abstract that British children were not included for inhibitory control task and standardised delay choice task (line 53-54). In the methods section, we stated that the British dataset was collected first to be utilized in Miller et al. (2019) comparative study of delay of gratification in children and New Caledonian crows (284-286). We also explained why we were unable to recruit the same group of British children and administer the inhibition and delay choice tasks (line 288-293). 

The overarching aim of the current study was to compare pre-schoolers’ delay of gratification in Britain and China and to investigate the role of reward representation on children’s self-control performance. Therefore, we think the introduction should focus on these points and not emphasize any cultural group. 

(1b) There were some instances where expressions were non-idiomatic or sounded somewhat awkward (“..need to maintain their action on the face of a tempting treat…”, “children of 4 years-old and above”, “..Chinese show advantageous performance on self-control tasks than their..”, etc.). There were further instances throughout the manuscript, and I think the manuscript would benefit from careful proof-reading and language editing.

The revised manuscript has been fully proof-read and we have corrected typos. 

(2) Methods: The methods are clearly described.

(2a) Participants: Further details on Chinese participants/data collection would be helpful. For example, when was the Chinese data collected? How many Chinese children per age group were tested for the study (currently, only numbers for British and Chinese children together are provided)?

We have incorporated more information on the Chinese participants and data collection in the methods section (line 278-283, line 286-288). 

(2b) The authors describe their counterbalancing within Experiments; was the order of Experiments also counterbalanced or did they take place in a fixed order?

All participants completed the study in a fixed order, Experiment 1 first followed by Experiment 2 (line 407-409). 

(2c) Delay of gratification task: The authors stated that the delayed reward remained inaccessible until the end of the study. How long was the delay and was the delay standardized across children (or did it vary)?

The delayed reward remained inaccessible until the end of the study. Children waited approximately 2 minutes and the delay length was same for all participants (446-448). 

(2d) Interrater reliability: usually Cohen’s Kappa or interrater-reliability coefficients (ICC) are reported as measures of reliability and not just percentage agreement. It would be helpful to add one of these standard measures.

Cohen’s Kappa was run as measures of reliability and there was good agreement across the raters, κ = .828, p<.001 (line 466-467). 

(3) Analyses: Using GLMMs to analyse the data is an adequate choice and full-null model comparisons are helpful to ensure the variables of interest predict outcome measures.

(3a) It would be good if the authors could add further details on their analytical approach: for example, which control variables did the null models contain? What function did they use for likelihood ratio tests to determine p-values of model predictors? For transparency, it would be helpful if the authors published their R code alongside their paper or, as a minimum, provide details on their model-formulas in the manuscript.

The null models contained the random effects and control variables (i.e. no predictor variables) namely “sex” as we did not predict this variable to significantly effect performance. The reduced models comprised of all effects present in the full model, except the effect of interest. For the GLMMs, we used family = binomial, R package “lme4”, “glmer” and “anova” functions (R version 3.4.3; Bates et al., 2015). We compared the log likelihood ratio of the a) full with null model, and b) full with reduced models to test each of the effects of interest, using maximum likelihood. We have added this information to the main text (lines 483-492). Additionally, we include a copy of our R script for the GLMM on Figshare: (https://figshare.com/s/01356d6162a8b55137c4) (line 492-493).

(3b) Furthermore, the authors may consider comparing a full model (with interaction term) with a reduced model (only main effects) to find the model with the best fit to the data, and then report the best fit model and its results.

For experiment 1 and 2, we compared the full model (with interaction term) with a reduced model containing only the main effects as suggested. In experiment 1, the full model was not significantly different to the reduced model (X2 = 3.34, df = 1, p = 0.067). Therefore, the interaction term (Age: Country) does not significantly improve the model and the reduced model is the best fit. We have updated our results table accordingly (Table 1) (line 519-523). In experiment 2, the full model is significantly different to the reduced model and is the best fit model (X2 = 15.361, df = 3, p = 0.002). We have added this information to the main text (line 552-553).

(3c) What really puzzled me (and, in fact, seemed redundant) are the non-parametric follow-up tests. The GLMMs already provide all the relevant information (i.e., is there a significant effect of country, condition, age, etc.). There are also packages available in R for follow-up tests for predictors with more than 2 levels (e.g., the lsmeans package). 

We agree that the GLMMS provide the most relevant information, i.e. which effects are significant. We then use this information to determine which variables to run post-hoc tests with to allow us to test the direction of these effects e.g. comparing performance at age 3 with age 4, age 4 with age 5, age 3 with age 5. We agree that different packages could be used in R for follow-up tests, but feel that our selection of non-parametric tests is also suitable, particularly as is then comparable analysis with our previous related study (Miller et al., 2019. Animal Cognition). 

Also, while age is entered as continuous variable in the models, it is then used as a categorical variable in the follow up tests. If the authors were interested in differences between age groups, they could have entered age as categorical variable in their GLMMs.

Apologies, this is our mistake in the text – we used age group as a categorical variable in the GLMMS (like the follow-up tests). We have now corrected this in the text (line 477).

(4) Results:

(4a) The studies also included preference and control trials, but the data are currently not reported. I think it would be helpful to report these data in the results section.

We have included the analyses in the supporting information for the test and control trials combined for Experiment 1 & 2. As trial type (test vs control) was significant in both models per experiment, and the test trials were our main focus, we present the test trial results in the main manuscript and now refer to the combined model output in the supporting information (line 546-547, 557-558, S1 Table, S2 Table). 

We have added data of the preference test for each country in the text (line 458-462). 

(4b) Table 2: it would be helpful to clarify that the p-values were derived from likelihood ratio tests.

Yes, the p-values are derived from the likelihood ratio tests – we now add this to the data analyses section for clarity (line 491-492). 

(4c) Please also see previous comments that it is not clear why non-parametric tests are reported.

For the data on Bramlett et al. (2012) rotating tray task, please see above Question (3c) for our justifications of using the non-parametric tests. For the inhibitory control tasks, the data from Chinese preschoolers were not normally-distributed thus did not meet the underlying assumptions for parametric tests (ANOVA and partial and Pearson’s correlation) (line 504-507). We have reported data from similar tasks and used the same non-parametric tests (Miller et al., 2019). 

(5) The discussion elaborates on the findings in detail. It could have been more concisely written at times, but I leave it up to the authors whether they would want to edit the discussion section.

Thanks for the suggestion of a more concise discussion. We have shortened the discussion meanwhile incorporated several points suggested by Reviewer 1. We feel the discussion section in its present form is informative and explains our findings with an appropriate level of detail.

Minor comment:

1st paragraph, introduction: “From financial decision in humans to foraging behaviours in other animals, we frequently face intertemporal choices in which they weigh the costs and and benefits…” This sentence sounds as if “we” are also “other animals” and switches somewhat awkwardly between “we” and “they”. Reword?

We have reworded the sentence to avoid any confusion (line 74).

For future submissions, it would be helpful if the manuscript file included page numbers, so they can be used as reference for comments.

We have included page numbers and line numbers in the revised manuscript and ensure that the format is in line with PLOS ONE’s guidelines.

---

## [Decision Letter · Decision Letter 1]

11 Jul 2021

PONE-D-21-06290R1

Waiting for the Better Reward: Comparison of Delay of Gratification in Young Children across Two Cultures

PLOS ONE

Dear Dr. Miller (Harrison),

Thank you for submitting your manuscript to PLOS ONE. After careful consideration, we feel that it has merit but does not fully meet PLOS ONE’s publication criteria as it currently stands. Therefore, we invite you to submit a revised version of the manuscript that addresses the points raised during the review process.

In particular, please carefully address the issues raised by Reviewer 2 concerning statistical analyses and report also in the abstract, introduction, results and discussion that the data on British children have already been collected for a previous study. 

We look forward to receiving your revised manuscript.

Kind regards,

Elsa Addessi

Academic Editor

PLOS ONE

Journal Requirements:

Reviewers' comments:

Reviewer's Responses to Questions

**Comments to the Author**

1. If the authors have adequately addressed your comments raised in a previous round of review and you feel that this manuscript is now acceptable for publication, you may indicate that here to bypass the “Comments to the Author” section, enter your conflict of interest statement in the “Confidential to Editor” section, and submit your "Accept" recommendation.

Reviewer #1: All comments have been addressed

Reviewer #2: (No Response)

2. Is the manuscript technically sound, and do the data support the conclusions?

Reviewer #1: Yes

Reviewer #2: Partly

3. Has the statistical analysis been performed appropriately and rigorously? 

Reviewer #1: Yes

Reviewer #2: I Don't Know

4. Have the authors made all data underlying the findings in their manuscript fully available?

Reviewer #1: Yes

Reviewer #2: Yes

5. Is the manuscript presented in an intelligible fashion and written in standard English?

Reviewer #1: Yes

Reviewer #2: Yes

6. Review Comments to the Author

Reviewer #1: I appreciate the authors' responsiveness to the reviewer feedback and believe that they have adequately addressed the comments from the previous round for an improved and interesting manuscript.

As a very minor comment, when referencing the studies (47-52), on lines 153-154 and 702-704, be sure to be inclusive of the pig reference (right now, the authors refer to these studies as primates and avian species but study 51 includes pigs).

Reviewer #2: The authors have considerably revised their manuscript and I found the overall manuscript improved. The authors have addressed the majority of reviewer comments, however, there remain a few issues that I think would need to be addressed and/or clarified in a further revision.

(1) Use of previously collected data:

I commented in my initial review that the fact that the British data originates from a previously published study did not become clear until the methods section and recommended to centre the paper more on the novel Chinese data. The authors responded that the comparison of the two data sets is central to their paper and that they did not want to emphasize one group over the other. I have no objections to this point (it is up to the authors to decide on the emphasis of their paper); however, the authors still do not clearly state in the abstract or in the introduction that the only novel data are the Chinese data. In fact, statements in the abstract like “Here, we tested delay of gratification in 136 3 to 5-year-old British (n=61) and Chinese (n=75) children using Bramlett et al. (1) delay choice paradigm…” give the impression that all the presented data were novel. For transparency reasons, I think it is important to clearly state throughout the manuscript that the British data used for comparison has previously been published (and reference the respective publication). I think this needs to be stated (and the publication referenced) throughout including the abstract, introduction, figure captions, table headings and discussion section, so even readers how may skip sections of the paper are aware of this.

(2) Statistical analyses:

In my previous comments, I asked the authors for some clarifications about their statistical approach. They have now provided further details and helpfully provided examples from their Rscript (their data is also available for download from figshare). However, I have some remaining comments/questions about the analyses.

(2a) P-values of main effects in the best fit model: Unfortunately, based on the Rscript provided it still did not fully become clear how the p-values were determined. It seems like the authors manually removed each factor of interest and then compared models, but the full script is not provided (only what seems like an example is provided). [Note: likelihood ratio tests for all factors can be quickly done with the drop1() function, using the best-fit model as input and setting the test statistic to chi-square; e.g. drop1(m4, test=”Chisq”).]

On my second reading of the paper, I also noticed that the model tables (Table 1 and 2) report z-values for the estimates. If the reported p-values are derived from likelihood ratio tests, then the reported test statistic would need to be chi-square values and dfs from those tests. The z-values refer to approximate p-values from the model outputs and do not correspond to the likelihood ratio test statistic (which is a chi-square test statistic).

(2b) Age as continuous vs. categorical variable in the GLMMs: The authors now clarified that age was entered as categorical variable in the GLMMs. However, the R-script does not convert age into a factor before analysis and age is a numerical variable in the data file. So based on this information, it seems that age may have been entered as a continuous variable? Could the authors double check and, if necessary, re-run their analyses with age as categorical variable?

(2c) Additional non-parametric post-hoc tests: I commented in my previous review that I am puzzled about the non-parametric follow-up tests and the authors offered two reasons for conducting them (a) “to test the direction of these effects” and (b) “comparable analysis with our previous related study (Miller et al., 2019, Animal Cognition”.

About (a): the model estimates (i.e., their sign) provide information about the direction of the effects; for example, the estimate of condition in experiment 1 is negative and assuming that quality was set as reference level, this reveals that children’s success rate was lower in the quantity as compared to the quality condition. For any factor with two levels (which are the majority of the authors’ factors), the initial model output is sufficient to interpret the direction of the effects. For factors with more than 2 levels (such as age), the authors could set the reference level of the factor to e.g. 3yos using the relevel() function and will then get estimates for 3 vs. 4 and 3 vs. 5; if they also want an estimate of 4 vs. 5 they can change the reference level to 4yos and re-run the model.

About (b): I had a look at the methods and results section of the previous paper (Miller et al., 2019). The paper reports model comparisons and tables with model estimates and p-values (derived from, what I assume, are likelihood ratio tests). Non-parametric tests were only reported for comparisons against chance; no non-parametric test statistics for e.g. comparing the two conditions were reported. So, it seems that the authors directly interpreted the model estimates in their previous article (?) and only used non-parametric tests for comparisons against chance. (Note that my puzzlement only concerns the use of non-parametric effects to interpret the model effects, not their use for comparisons against chance.)

(3) Discussion, p. 21, line 668 ff.: This paragraph makes some broad claims about country-level differences in performance although the data is more nuanced than that. In fact, the authors have a very insightful and nuanced discussion of cultural similarities and differences later in the discussion section and I think the later section does better justice to their data. So, I would recommend removing the paragraph in lines 668-678.

Minor comment:

The authors frequently use wording like "Bramlett et al. (1) delay choice paradigm" when referencing paradigms or previous findings. I think this may need to be changed to "Bramlett et al.'s (1) delay choice paradigm" etc..

7. PLOS authors have the option to publish the peer review history of their article (what does this mean?). If published, this will include your full peer review and any attached files.

Reviewer #1: No

Reviewer #2: **Yes: **Patricia Kanngiesser

---

## [Author Response · Author response to Decision Letter 1]

25 Jul 2021

Dear Dr Addessi,

Many thanks for inviting us to revise our manuscript [PONE-D-21-06290R1] entitled “Waiting for the Better Reward: Comparison of Delay of Gratification in Young Children across Two Cultures”. 

We wish to thank you and the two reviewers for helpful feedback. We have now addressed the comments. Please find responses to each comment below. Please note that line numbers correspond with the tracked changes version of the manuscript.

We hope that following our revisions, you will consider our manuscript for publication in PLoS ONE.

Yours Sincerely.

Ning Ding, Anna Frohnwieser, Rachael Miller, and Nicky Clayton

Response to Editor and Reviewers:

Editor comment:

In particular, please carefully address the issues raised by Reviewer 2 concerning statistical analyses and report also in the abstract, introduction, results and discussion that the data on British children have already been collected for a previous study. 

We have now addressed these comments in the ms and in response to reviewers below. 

Reviewer comments:

Reviewer #1: I appreciate the authors' responsiveness to the reviewer feedback and believe that they have adequately addressed the comments from the previous round for an improved and interesting manuscript.

Thank you very much.

As a very minor comment, when referencing the studies (47-52), on lines 153-154 and 702-704, be sure to be inclusive of the pig reference (right now, the authors refer to these studies as primates and avian species but study 51 includes pigs).

We have changed the text to state that the research includes primates, pigs, and avian species (line 141-142, line 703). 

Reviewer #2: The authors have considerably revised their manuscript and I found the overall manuscript improved. The authors have addressed the majority of reviewer comments, however, there remain a few issues that I think would need to be addressed and/or clarified in a further revision.

Thank you very much, we have now addressed these additional comments. 

(1) Use of previously collected data:

I commented in my initial review that the fact that the British data originates from a previously published study did not become clear until the methods section and recommended to centre the paper more on the novel Chinese data. The authors responded that the comparison of the two data sets is central to their paper and that they did not want to emphasize one group over the other. I have no objections to this point (it is up to the authors to decide on the emphasis of their paper); however, the authors still do not clearly state in the abstract or in the introduction that the only novel data are the Chinese data. In fact, statements in the abstract like “Here, we tested delay of gratification in 136 3 to 5-year-old British (n=61) and Chinese (n=75) children using Bramlett et al. (1) delay choice paradigm…” give the impression that all the presented data were novel. For transparency reasons, I think it is important to clearly state throughout the manuscript that the British data used for comparison has previously been published (and reference the respective publication). I think this needs to be stated (and the publication referenced) throughout including the abstract, introduction, figure captions, table headings and discussion section, so even readers how may skip sections of the paper are aware of this.

Thank you for pointing out the need to state the use of previously collected data and it is now explicitly stated in the Abstract (line 37), Introduction (line 233-235), figure captions (line 531-532, line 538, line 588), table legends (line 527, line 582) and discussion (line 633-635). 

(2) Statistical analyses:

In my previous comments, I asked the authors for some clarifications about their statistical approach. They have now provided further details and helpfully provided examples from their Rscript (their data is also available for download from figshare). However, I have some remaining comments/questions about the analyses.

(2a) P-values of main effects in the best fit model: Unfortunately, based on the Rscript provided it still did not fully become clear how the p-values were determined. It seems like the authors manually removed each factor of interest and then compared models, but the full script is not provided (only what seems like an example is provided). [Note: likelihood ratio tests for all factors can be quickly done with the drop1() function, using the best-fit model as input and setting the test statistic to chi-square; e.g. drop1(m4, test=”Chisq”).]

On my second reading of the paper, I also noticed that the model tables (Table 1 and 2) report z-values for the estimates. If the reported p-values are derived from likelihood ratio tests, then the reported test statistic would need to be chi-square values and dfs from those tests. The z-values refer to approximate p-values from the model outputs and do not correspond to the likelihood ratio test statistic (which is a chi-square test statistic).

We hope we have improved clarity of the R script (https://figshare.com/s/01356d6162a8b55137c4) and updated the GLMM results in the manuscript and supplementary materials to include chi-square values and dfs rather than z-values for estimates (Table 1, Table 2, S1, S2, S3). The p-values have been checked and updated if required and reflect those obtained from the likelihood ratio tests.

(2b) Age as continuous vs. categorical variable in the GLMMs: The authors now clarified that age was entered as categorical variable in the GLMMs. However, the R-script does not convert age into a factor before analysis and age is a numerical variable in the data file. So based on this information, it seems that age may have been entered as a continuous variable? Could the authors double check and, if necessary, re-run their analyses with age as categorical variable?

We have re-run our analyses ensuring that age was converted into a factor before analysis (i.e. as a categorical variable). There are no changes for Experiment 1 (Fig 4 updated to better represent the overall age effect, line 534-538). In Experiment 2, age and country are no longer significant main effects, while the country: age interaction and condition remains significant. We have updated the results section accordingly (Table 2, line 558-559, line 564-577) and modified Fig 5 (line 584-588) to present the interaction effect. We also changed the supplementary tables (outliers removed) and confirm again that there are no significant changes to the results from Experiment 2 when the outliers are removed. 

(2c) Additional non-parametric post-hoc tests: I commented in my previous review that I am puzzled about the non-parametric follow-up tests and the authors offered two reasons for conducting them (a) “to test the direction of these effects” and (b) “comparable analysis with our previous related study (Miller et al., 2019, Animal Cognition”.

About (a): the model estimates (i.e., their sign) provide information about the direction of the effects; for example, the estimate of condition in experiment 1 is negative and assuming that quality was set as reference level, this reveals that children’s success rate was lower in the quantity as compared to the quality condition. For any factor with two levels (which are the majority of the authors’ factors), the initial model output is sufficient to interpret the direction of the effects. For factors with more than 2 levels (such as age), the authors could set the reference level of the factor to e.g. 3yos using the relevel() function and will then get estimates for 3 vs. 4 and 3 vs. 5; if they also want an estimate of 4 vs. 5 they can change the reference level to 4yos and re-run the model.

About (b): I had a look at the methods and results section of the previous paper (Miller et al., 2019). The paper reports model comparisons and tables with model estimates and p-values (derived from, what I assume, are likelihood ratio tests). Non-parametric tests were only reported for comparisons against chance; no non-parametric test statistics for e.g. comparing the two conditions were reported. So, it seems that the authors directly interpreted the model estimates in their previous article (?) and only used non-parametric tests for comparisons against chance. (Note that my puzzlement only concerns the use of non-parametric effects to interpret the model effects, not their use for comparisons against chance.)

We have followed the reviewer’s suggestion and now directly interpret the model estimates rather than reporting non-parametric test results for within-factor comparisons (e.g. sex, country, order, condition). We also now use Tukey contrasts to report pair-wise comparisons within the (3-level) factor of age where applicable and updated the manuscript accordingly (line 542-545; line 566-569; line 572-575). We continue to use non-parametric tests to report comparisons against chance, which the reviewer supported in their comment. 

(3) Discussion, p. 21, line 668 ff.: This paragraph makes some broad claims about country-level differences in performance although the data is more nuanced than that. In fact, the authors have a very insightful and nuanced discussion of cultural similarities and differences later in the discussion section and I think the later section does better justice to their data. So, I would recommend removing the paragraph in lines 668-678.

We are very glad that the reviewer considered our discussion insightful and nuanced. We have included the content about societal difference between the Eastern and Western as a broader influence of children’s cognitive development and it also echoes the literature we summarised in the introduction. We think it’s worth keeping it because it might help with readers who haven’t read the introduction or with limited knowledge of cross-cultural research. 

Minor comment:

The authors frequently use wording like "Bramlett et al. (1) delay choice paradigm" when referencing paradigms or previous findings. I think this may need to be changed to "Bramlett et al.'s (1) delay choice paradigm" etc..

We have changed the reference throughout the manuscript.

---

## [Decision Letter · Decision Letter 2]

20 Aug 2021

Waiting for the Better Reward: Comparison of Delay of Gratification in Young Children across Two Cultures

PONE-D-21-06290R2

Dear Dr. Miller (Harrison),

We’re pleased to inform you that your manuscript has been judged scientifically suitable for publication and will be formally accepted for publication once it meets all outstanding technical requirements.

Kind regards,

Elsa Addessi

Academic Editor

PLOS ONE

Additional Editor Comments (optional):

Reviewers' comments:

Reviewer's Responses to Questions

**Comments to the Author**

1. If the authors have adequately addressed your comments raised in a previous round of review and you feel that this manuscript is now acceptable for publication, you may indicate that here to bypass the “Comments to the Author” section, enter your conflict of interest statement in the “Confidential to Editor” section, and submit your "Accept" recommendation.

Reviewer #2: All comments have been addressed

2. Is the manuscript technically sound, and do the data support the conclusions?

Reviewer #2: Yes

3. Has the statistical analysis been performed appropriately and rigorously? 

Reviewer #2: Yes

4. Have the authors made all data underlying the findings in their manuscript fully available?

Reviewer #2: Yes

5. Is the manuscript presented in an intelligible fashion and written in standard English?

Reviewer #2: Yes

6. Review Comments to the Author

Reviewer #2: The authors have addressed all comments. Congratulations on this interesting study and fine manuscipt!

7. PLOS authors have the option to publish the peer review history of their article (what does this mean?). If published, this will include your full peer review and any attached files.

Reviewer #2: **Yes: **Patricia Kanngiesser

---

## [Editor Report · Acceptance letter]

27 Aug 2021

PONE-D-21-06290R2 

Waiting for the better reward: Comparison of delay of gratification in young children across two cultures 

Dear Dr. Miller (Harrison):

I'm pleased to inform you that your manuscript has been deemed suitable for publication in PLOS ONE. Congratulations! Your manuscript is now with our production department. 

Kind regards, 

on behalf of

Dr. Elsa Addessi 

Academic Editor

PLOS ONE